# Neuromodulatory subcortical nucleus integrity is associated with white matter microstructure, tauopathy and *APOE* status

Alfie Wearn ®[1] ✉, Stéfanie A. Tremblay[2,3,4], Christine L. Tardif[1,5,6], Ilana R. Leppert[1,5], Claudine J. Gauthier[2,3,4], Giulia Baracchini ®[1], Colleen Hughes[1], Patrick Hewan[7], Jennifer Tremblay-Mercier[8], Pedro Rosa-Neto ®[1,5,8], Judes Poirier ®[8,9], Sylvia Villeneuve ®[5,8,9], Taylor W. Schmitz ®[10], Gary R. Turner ®[7], R. Nathan Spreng ®[1,5,8,9] ✉ & PREVENT-AD Research Group*

The neuromodulatory subcortical nuclei within the isodendritic core (IdC) are the earliest sites of tauopathy in Alzheimer's disease (AD). They project broadly throughout the brain's white matter. We investigated the relationship between IdC microstructure and whole-brain white matter microstructure to better understand early neuropathological changes in AD. Using multi-parametric quantitative magnetic resonance imaging we observed two covariance patterns between IdC and white matter microstructure in 133 cognitively unimpaired older adults (age 67.9 ± 5.3 years) with familial risk for AD. IdC integrity related to 1) whole-brain neurite density, and 2) neurite orientation dispersion in white matter tracts known to be affected early in AD. Pattern 2 was associated with CSF concentration of phosphorylated-tau, indicating AD specificity. Apolipoprotein-E4 carriers expressed both patterns more strongly than non-carriers. IdC microstructure variation is reflected in white matter, particularly in AD-affected tracts, highlighting an early mechanism of pathological development.

Healthy brain function relies on the isodendritic core (IdC), a group of neuromodulatory nuclei with diffuse projections throughout the entire brain[1–4]. The IdC comprises the noradrenergic locus coeruleus (LC), serotonergic dorsal raphe (DR) (both in the brainstem), dopaminergic ventral tegmental area (VTA) (in the midbrain), and the cholinergic nucleus basalis of Meynert (NbM) in the basal forebrain. Due to their extremely large physical size and correspondingly high demand for resources, their neurons are especially vulnerable

to damage in age and disease when these demands can no longer be met[5–7].

IdC nuclei comprise some of the earliest known sites of tauopathy in Alzheimer's disease (AD), the leading cause of dementia worldwide. Tauopathy is evident in the IdC decades before cognitive decline, and prior to degeneration of cortical regions including entorhinal cortex, as revealed by histopathological[8–17] and neuroimaging studies[9,18–27]. Braak staging describes pretangle stages a–c, characterized by IdC

[1]Department of Neurology and Neurosurgery, Montreal Neurological Institute, McGill University, Montreal H3A 2B4 QC, Canada. [2]Department of Physics, Concordia University, Montreal H4B 1R6 QC, Canada. [3]Montreal Heart Institute, Montreal H1T 1C8 QC, Canada. [4]School of Health, Concordia University, Montreal H4B 1R6 QC, Canada. [5]McConnell Brain Imaging Centre, McGill University, Montreal H3A 2B4 QC, Canada. [6]Department of Biomedical Engineering, McGill University, McGill H3A 2B4 QC, Canada. [7]Department of Psychology, York University, Toronto M3J 1P3 ON, Canada. [8]Douglas Mental Health University Institute—Research Center, Verdun H4H 1R3 QC, Canada. [9]Department of Psychiatry, McGill University, Montreal H3A 1A1 QC, Canada. [10]Department of Physiology & Pharmacology, Western Institute for Neuroscience, Western University, London N6A 5C1 ON, Canada. *A list of authors and their affiliations appears at the end of the paper. ✉e-mail: alfie.wearn@mcgill.ca; nathan.spreng@mcgill.ca

## BOX 1

# Interpretation of measures of microstructure in the context of AD

Advances in quantitative magnetic resonance imaging (MRI) allow for unprecedented in vivo characterization of human aging and brain health. Multiparametric mapping (MPM) protocols provide complementary measures that enable comprehensive in vivo assessment of tissue composition and microstructure. Each measure is differentially sensitive to different elements of tissue microstructure (Box Table 1). Neurite Orientation Dispersion and Density Imaging (NODDI) improves the specificity of diffusion imaging-derived measures by leveraging multi-shell diffusion encoding and a multi-compartment biophysical model of brain tissue (Box Table 2).

Box Table 1 | **Interpretation of MPM measures in the context of AD**.

| MPM parameter | Microstructural sensitivities | Expected effect of AD |
|---|---|---|
| R1 (1/T1) | Physicochemical environment. macro-molecular content (e.g. myelin (+), iron (+), water (−). | R1 decreases as neurites atrophy and tissue water increases. Conversely, increased iron will increase R1, so R1 tends to increase in iron-rich regions[116]. |
| MTsat (magnetization transfer saturation) | Macromolecular content (e.g. myelin) (++), iron (+), water (−). | MTsat decreases as myelination decreases and neurites atrophy[44,82]. |
| R2* (1/T2*) | Paramagnetic and diamagnetic materials, e.g. Iron (++), myelin (+), water (−). | R2* increases with increasing neurotoxic iron deposits but decreases with myelin dysregulation. Which factor influences signal the most varies by region and tissue type[117]. |
| PD (proton density) | MRI-visible water (+). | PD increases as CSF in voxels increases due to tissue atrophy or inflammatory edema. AD-related decreases in PD have been previously reported in basal forebrain[118]. |

For each measure, (+) and (−) indicate positive and negative associations with a given material, respectively. (++) represents a particularly strong driving factor. For a more detailed summary of the sensitivities of these measures see references[40,42,82,116].
*MPM* multiparametric mapping.

Box Table 2 | **Interpretation of NODDI measures in the context of AD**.

| NODDI parameter | Microstructural sensitivities | Expected effect of AD |
|---|---|---|
| NDI (neurite density index) | Packing density of neurites. | Atrophy or shrinking of neurites causes lower packing density and lower NDI[51,65]. |
| ODI (orientation dispersion index) | Dispersion of neurites (sensitive to the presence of crossing fibers). | Selective sparing or degeneration of crossing fibers in a given bundle causes increase or decrease of ODI, respectively. Measures of ODI in response to neurodegeneration therefore vary by region and disease stage[51,65,73,74]. |
| FW (free water fraction) | Free water is not associated with neurites. | As cells atrophy, less water is associated with neurites and more water is 'free', causing FW to increase[65,74]. |

For a more detailed summary of the sensitivities of these measures see refs. 51,52.
*NODDI* neurite orientation dispersion and density imaging.

tauopathy, prior to the stage I neurofibrillary tangle formation[12]. The long-reaching and highly arborized axonal projections of the IdC that enable broad control of healthy brain function are an efficient mechanism of tau propagation throughout the brain under the trans-synaptic spread model[28,29].

Selective damage to the neurons within IdC nuclei can have significant, wide-ranging consequences for health and behavior due to their diffuse projections across the entire brain[5]. Abnormalities within the IdC cause dysfunction across a range of behaviors, including mood[17,30–32], sleep cycles[33,34] memory[35], and attention[36,37]. Such symptoms are commonly reported in the prodromal phase of AD but are difficult to objectively assess or dissociate from inter-individual differences in trajectories of healthy aging in the early stages of the disease[38]. IdC integrity may, therefore, constitute a trans-symptom indicator of global brain health. Given the wide spatial distribution and highly arborized axonal structure of IdC neurons, we predicted that measures of distributed white matter integrity across the entire brain would reflect disruption of the local integrity of IdC nuclei.

Advances in quantitative magnetic resonance imaging (MRI) allow for unprecedented in vivo characterization of human aging and brain

health. The ability to derive quantitative units from MRI scans greatly improves the biological interpretability and consistency of the MR signal[39–41]. High-resolution multiparametric mapping (MPM) protocols[40] provide multiple complementary measures that enable comprehensive in vivo assessment of IdC composition and microstructure. In this study, we use MPM to derive four such parameters from a single session of MRI: R1 (1/T1), magnetization transfer saturation (MTsat), R2* (1/T2*), and proton density (PD). R1 represents the longitudinal relaxation rate and is sensitive to a combination of factors, including macromolecular content, water, and paramagnetic materials[42,43]. MTsat is sensitive to the macromolecular content of a voxel and is driven largely by lipids and proteins within the phospholipid bilayer of cells (especially myelin), as well as macromolecular protein depositions[42,44]. R2* is related to the deposition of iron that can cause damage through reaction with reactive oxygen species, as well as the diamagnetic effects of myelin[45,46]. PD measures the concentration of protons in a tissue and, therefore, broadly represents water content, which can increase due to edema or cell death[47]. Further information about these measures and their expected changes due to AD can be found in Box 1.

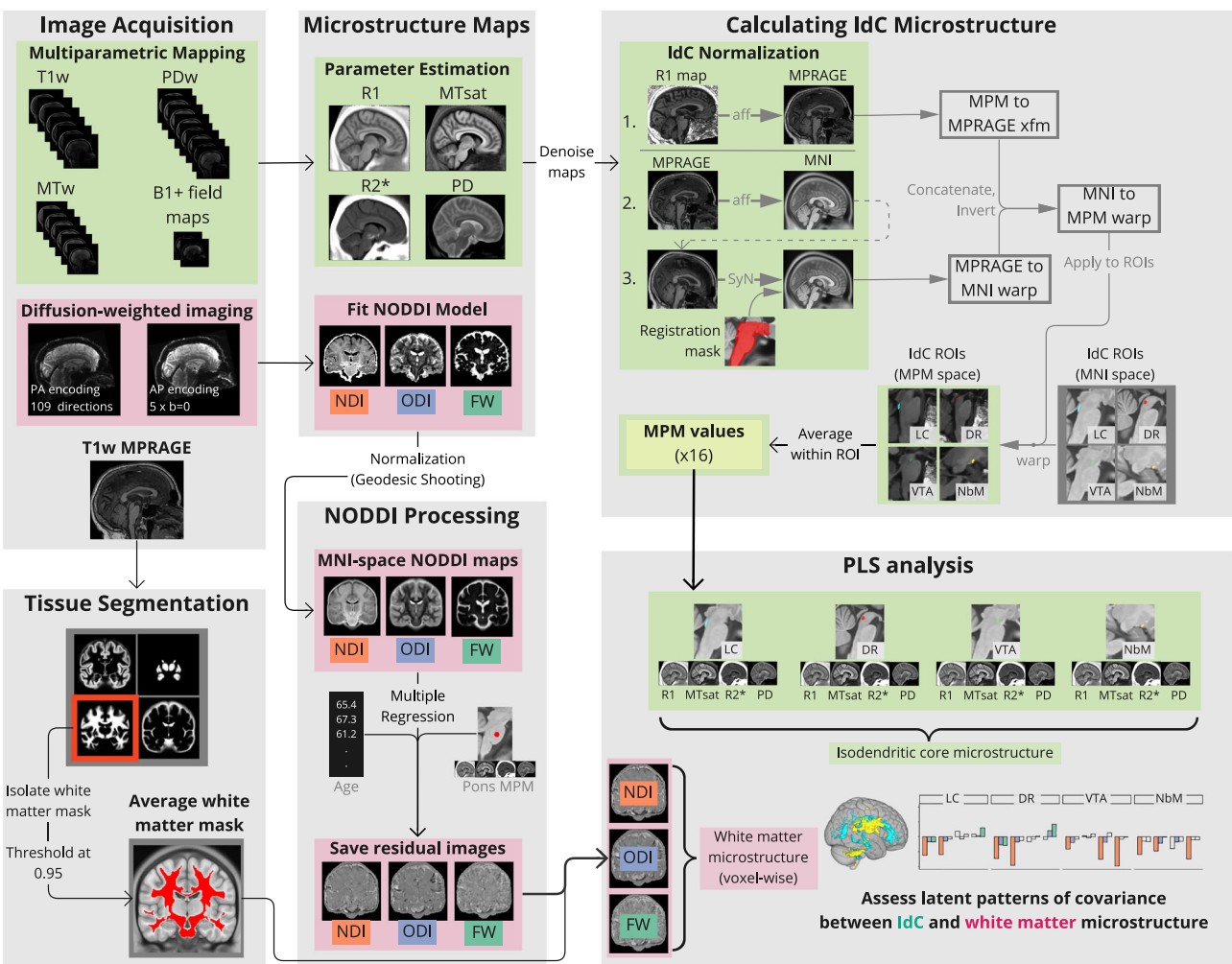

**Fig. 1 | Flowchart of image processing pipeline.** Major parts of the pipeline from image acquisition to the primary (partial least squares) analysis of this study are shown, with major sections shown with a light gray background. Green boxes refer to parts of the multiparametric-mapping (MPM) processing pipeline. Pink boxes refer to parts of the diffusion processing pipeline. Smaller intermediary processes (e.g. denoising) are shown with text along arrows and intermediary products (e.g. transform matrices) are shown in a dark gray box. Images for all masks are shown in more detail in Supplemental Information. MPM multiparametric mapping, MTsat magnetization transfer saturation, PD proton density, NODDI neurite orientation dispersion and density imaging, NDI neurite density index, ODI orientation dispersion index, FW free water fraction, IdC isodendritic core, LC locus coeruleus, DR dorsal raphe, VTA ventral tegmental area, NbM nucleus basalis of Meynert, PLS partial least squares.

Complementing these MPM metrics, diffusion-weighted imaging estimates of white matter microstructure related to axonal damage, neuronal loss, and inflammation (key features of early Alzheimer's pathology)[48–50] can now also be measured with improved sensitivity and specificity using the neurite orientation dispersion and density imaging (NODDI) model[51,52]. Multiple microstructural features can be derived from NODDI, including: neurite density index (NDI), orientation dispersion index (ODI), and free-water fraction (FW)[51,52]. NDI represents the overall density of axons and dendrites within a voxel. ODI represents the variety of directions (the 'dispersion') exhibited by water within and around these neurites, where values between 0 and 1 represent perfectly aligned to maximally (isotropically) dispersed neurites, respectively. Finally, FW represents the isotropic free-diffusion compartment—water which is neither within nor directly associated with neurites. Together, these measures can provide a more detailed view of the microstructure within each white matter voxel, allowing for greater interpretability as to the underlying mechanisms of signal variation in white matter[51,52]. Further information about these measures and their expected changes due to AD can be found in Box 1.

In this study, we investigated how the microstructure of IdC nuclei was associated with white matter microstructure across the whole brain in a cohort of cognitively unimpaired older adults with familial risk for AD. We used MPM-derived metrics (R1, MTsat, R2*, and PD) to quantify multiple microstructural features of the IdC and NODDI to quantify multiple microstructural features of whole-brain white matter. The multivariate covariance patterns captured by these features were then related to cerebrospinal fluid (CSF) biomarkers for AD. We additionally assessed the specificity of these patterns to Apolipoprotein-ε4 (*APOE4*) genotype, the strongest genetic risk factor for sporadic AD, given its role in axonal myelination regulation[53]. This work provides insight into the IdC projection system, the earliest targets of AD pathology in the presymptomatic stages of disease progression[1–3,12].

## Results

### Covariance patterns of white-matter and IdC microstructure

To investigate how microstructure of the IdC is associated with whole-brain white matter microstructure, we leveraged state-of-the-art structural neuroimaging techniques to calculate distinct metrics of microstructural integrity for IdC nuclei (R1, MTsat, R2*, PD) and whole-brain white matter (ODI, NDI, FW). The image processing pipeline and IdC ROI masks are shown in Fig. 1 and Supplementary Fig. 1, respectively.

**Table 1 | Average MPM values for all IdC nuclei and Pontine region of no interest**

| All subjects | LC | DR | VTA | NbM | Pontine region |
|---|---|---|---|---|---|
| R1 (1/s) | 0.61 ± 0.03 | 0.58 ± 0.03 | 0.64 ± 0.04 | 0.65 ± 0.03 | 0.79 ± 0.04 |
| MTsat (p.u.) | 0.90 ± 0.10 | 0.84 ± 0.09 | 0.93 ± 0.11 | 0.91 ± 0.07 | 1.44 ± 0.12 |
| R2* (1/s) | 15.0 ± 2.07 | 14.8 ± 1.75 | 19.0 ± 2.54 | 24.9 ± 2.48 | 20.2 ± 2.48 |
| PD (p.u.) | 76.1 ± 1.42 | 76.1 ± 1.38 | 76.6 ± 1.84 | 77.5 ± 1.28 | 68.8 ± 1.44 |

Values show mean ± standard deviation. Units are shown per parameter.
p.u. percent units.

We used partial least squares (PLS) analysis to determine multivariate patterns of covariance between IdC microstructure and whole brain white matter microstructure. PLS is a multivariate statistical approach that finds the maximal correlation structure between sets of variables of interest (i.e., MPM and NODDI measures in our case)[54]. With PLS, the correlation between MPM-derived IdC and voxelwise NODDI-derived white matter microstructure was decomposed into latent variables (LVs). Within each LV, each white matter voxel is assigned a singular value weight, a 'salience', for each NODDI measure, that is proportional to its correlation with the MPM-derived measures in the IdC ROIs. Calculating the dot-product of the salience weight with the NODDI value within that voxel gives a 'brain score' for each participant on each LV, which reflects the extent to which a given participant expresses the group pattern for that LV.

Mean MPM values for each IdC nucleus are shown in Table 1 (split by *APOE4* carrier status in Supplementary Table 2), and correlations with demographics and CSF Phosphorylated tau (threonine 181) (pTau181) and Amyloid-beta 1-42 (Aß42) are shown in Fig. 2.

We identified two significant LVs, representing correlation patterns of IdC integrity with (1) NDI and ODI across the whole-brain white matter and (2) localized variation in ODI.

### Pattern 1: IdC integrity is associated with whole-brain covariance of neurite density and orientation dispersion

The first significant LV explained 26.9% of total crossblock covariance ($p < 0.0001$) (Fig. 3). R1, MTsat and R2* across multiple IdC nuclei were positively associated with NDI and ODI across the entire brain's white matter, with the exceptions of the corpus callosum, cingulum bundle, and midbrain. Results are displayed in Fig. 3 and summarized for each IdC nucleus in detail below. This pattern was driven primarily by the association between IdC integrity and NDI in white matter (Fig. 3a).

In LC: R1 was positively associated with NDI, ODI and FW across almost all white matter voxels. MTsat was also positively associated with NDI and ODI. PD was negatively associated with FW across this region.

In DR: R1 was positively associated with NDI, ODI and FW. MTsat was also positively associated with NDI and ODI. PD was negatively associated with ODI and FW.

In VTA: R1 and R2* were both positively associated with NDI and ODI. PD was also positively associated with NDI.

In NbM: R1 was positively associated with NDI. MTsat was positively associated with both NDI and ODI. R2* was positively associated with ODI. PD was positively associated with NDI.

### Pattern 2: IdC integrity is associated with localized variation in orientation dispersion

The second significant LV explained 10.2% of crossblock covariance ($p = 0.002$) (Fig. 4). All measures of IdC integrity across all four IdC nuclei contributed to this pattern, showing associations with two primary white matter regions. The first white matter region includes the internal capsule, corona radiata, and cerebellar peduncles, hereafter referred to as brainstem-efferent tracts due to their common feature of strong direct connections from the brainstem. The second white matter region encompassed areas of the cingulum complex, including

the hippocampal cingulum, as well as the corpus callosum, which we refer to as limbic tracts. These two tract groups show an anticorrelated pattern of IdC parameters (displayed in Fig. 4 and summarized by IdC nucleus in detail below). Examination of behavioral LVs (Fig. 4a) reveals that this pattern is primarily driven by covariance of IdC nuclei with ODI, and, to a lesser extent, NDI in white matter in the opposite direction.

In LC: R1 and MTsat were both negatively associated with NDI and positively associated with ODI in the brainstem-efferent regions. R2* was positively associated with ODI in these regions. PD was negatively associated with ODI in these regions.

In DR: Associations between DR integrity and white matter were largely the same as those of LC. R1, MTsat and R2* were all negatively associated with NDI and positively associated with ODI in the brainstem-efferent regions. PD was negatively associated with ODI in these regions.

In VTA: R1 and MTsat were positively associated with ODI in the brainstem efferent region. R2* was positively associated with both ODI and FW in this region.

In NbM: R1 and MTsat were both negatively associated with NDI and positively associated with ODI and FW in the brainstem-efferent regions. R2* was positively associated with ODI and FW in these regions. PD was negatively associated with FW in these regions.

The described associations were all inverted in the limbic tracts.

### Pattern 2 is related to CSF pTau181

PLS identified two patterns relating the IdC to white matter integrity. The first was a global pattern, driven primarily by R1, wherein higher values were associated with greater NDI. The second pattern revealed R1, MTsat and R2* values across nuclei related to the ODI of specific tracts. The limbic tracts identified in this analysis have been observed to be vulnerable to Alzheimer's pathology[55–57]. For this reason, we assessed the association between the primary drivers of each of the two patterns (NDI for LV1, ODI for LV2) with CSF pTau181 concentration, controlling for sex and years of education. No association was observed with NDI in LV1 ($\beta = 0.146$, $t = 1.45$, $p_{adj} = 0.300$, Fig. 5a). However, we observed a significant association with ODI in LV2 ($\beta = 0.295$, $t = 2.93$, $p_{adj} = .008$, Fig. 5b), meaning the second pattern is more strongly expressed in people with a higher concentration of pTau181, suggesting this pattern may be specific to AD. In neither model did pTau181 concentration interact with days between CSF and MRI collection, so this was not included in the final models. We observed no significant relationship between brain scores and Aß42.

Also of note, we observed a significant positive relationship between NDI for LV1 and years of education ($\beta = 0.305$, $t = 3.01$, $p = 0.003$). There was no statistically significant effect of sex in either model.

### Covariance patterns are largely specific to *APOE4*+ group

Integrity of the IdC is more vulnerable in *APOE4* carriers[58], and the gene is essential for dendritic arborization and myelination[53,59]. For this reason, we predicted that the association between IdC nuclei, using MPM-derived measures, and white matter microstructure, using NODDI-derived measures, would depend on the presence of an *APOE4*

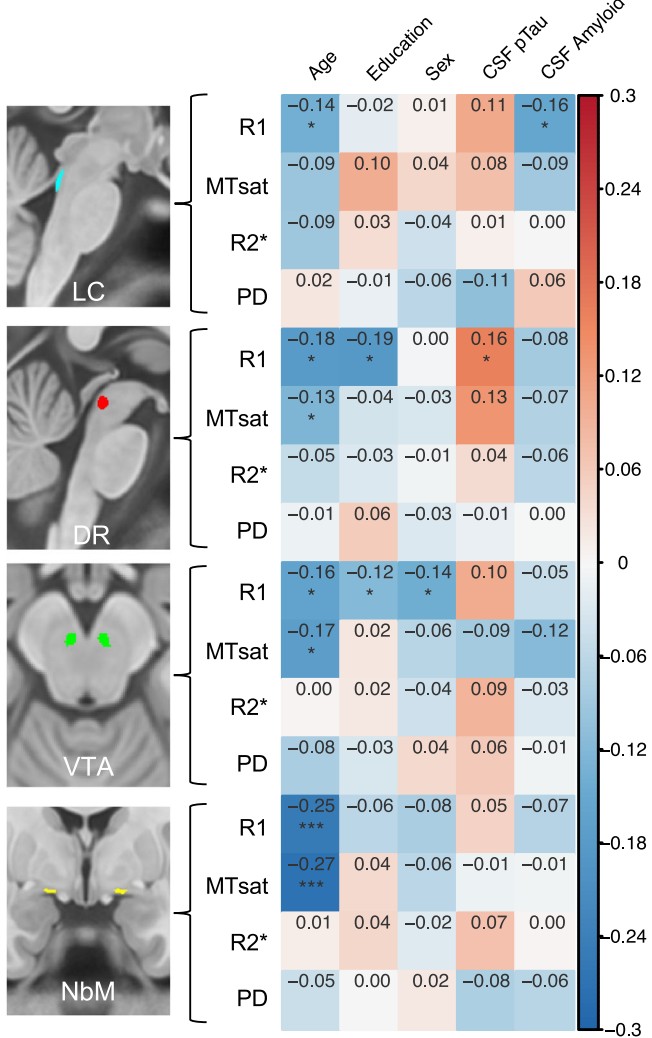

**Fig. 2 | Partial correlation statistics measures of Isodendritic core integrity and demographic variables and CSF biomarkers.** Correlation statistics are corrected for age, sex, and education. Sex was dummy-coded as male (1) and female (0), so a positive correlation indicates greater values in males, and a negative correlation indicates greater values in females. MT magnetization transfer saturation, PD proton density, LC locus coeruleus, DR dorsal raphe, VTA ventral tegmental area, NbM nucleus basalis of Meynert. *$p < 0.05$, **$p < 0.001$, ***$p < 0.0001$ (These $p$-values are uncorrected for multiple comparisons and are presented only for transparency. They are not the primary metrics interpreted in the analysis but are here only to guide future studies). All statistical tests shown were two-tailed. Source data are provided as a Source Data file.

allele. We repeated the same PLS analysis but split participants into *APOE4*+ ($n = 50$) and *APOE4*− ($n = 82$) groups. Four significant LVs emerged. The percentage of crossblock covariance explained by each pattern (and the significance of each LV) was: (1) 21.8% ($p < 0.0001$), (2) 8.42% ($p < 0.0001$), (3) 3.41%, ($p = 0.044$), and (4) 3.22% ($p = 0.022$). Full plots of all associations are displayed in Fig. 6, summarized briefly below, and described in full in supplementary information.

The pattern of significant voxels within white matter for LV1 and LV2 were highly similar to those identified in the primary analysis of the whole cohort and were expressed much more strongly in *APOE4* carriers than non-carriers. Therefore, in separating the cohort by genotype, it is clear that the primary results were driven by the *APOE4* carriers.

The third LV was largely specific to the *APOE4*- group. For LC and DR, higher values of R1, MTsat and R2* and lower PD were primarily associated with higher FW, but also higher NDI and ODI across brainstem-efferent tracts and peripheral posterior white matter. In

contrast, lower R1, MTsat and R2* and greater PD were associated with higher FW (as well as NDI and ODI) across limbic areas such as the cingulum and longitudinal fasciculus. This pattern was slightly more widespread in the right hemisphere.

The final significant pattern revealed a complex pattern of associations between IdC integrity and NODDI measures between *APOE* groups. The most striking feature was a group interaction in LC and DR between R1 and MTsat for ODI. In the *APOE4*+ group, greater R1 and MTsat values in LC and DR nuclei were associated with lower ODI in the superior longitudinal fasciculus and higher ODI in the corpus callosum and hippocampal cingulum. The reverse was observed in the *APOE4*− group. Here, we observed that greater R1 and MTsat values in LC and DR nuclei were associated with greater ODI in the superior longitudinal fasciculus and lower ODI in the corpus callosum and hippocampal cingulum. Also, in the *APOE4*− group, the NbM showed similar covariance as the LC and DR, with R1 and MTsat in association with ODI; VTA also covaried MTsat with ODI in the *APOE4*− group. Significant associations with FW were observed, which varied by MPM metric and group.

In line with the primary analysis, we also assessed the association between the primary drivers of each of the four patterns (Pattern 1: NDI, Pattern 2: ODI, Pattern 3: FW, Pattern 4: ODI) against CSF pTau181 concentration. We did not find evidence of a modulatory effect of *APOE4* on brain score associations with CSF pTau181 or Aβ42 for any LV. We did observe a similar relationship between CSF pTau181 and ODI in pattern 2, in line with the primary analysis, though it did not reach the threshold for statistical significance ($\beta = 3.13$, $t = 1.90$, $p = 0.06$) (Supplementary Fig. 3).

## Discussion

We have demonstrated a critical connection between the structure of the neuromodulatory nuclei of the IdC and whole-brain white matter microstructure. In a sample of cognitively unimpaired older adults at increased risk for AD, we employed quantitative MRI measures to determine the integrity of IdC nuclei, which are highly susceptible to Alzheimer's pathology in the earliest stages of the disease. We identified an association between IdC integrity and neurite density across the white matter, suggestive of a pattern of overall brain health. We also identified a second pattern relating IdC to the microstructural integrity of white matter limbic tracts and brainstem-efferent fibers. This second pattern was associated with CSF pTau181 concentration, suggesting specificity for AD. Both patterns were largely specific to *APOE4* carriers, with a non-carrier-specific pattern characterizing LC and DR covariance with localized free-water in white matter. *APOE4* carrier status also modulated the relationship of MTsat in LC and DR with localized changes in white matter. These findings indicate early pathological changes in presymptomatic AD progression.

### Interpretation of measures of IdC integrity

We included multiple microstructural measures in our analysis using robust multivariate statistical methods to create a comprehensive picture of what drives a given observation compared to choosing any single measure in isolation. In each pattern, white matter integrity was associated with a positive covariance of R1, MTsat and R2* within IdC. Within the LC and VTA, measures of MTsat and R2* may be directly sensitive to neuromelanin, a macromolecule with a strong affinity for lipids and a strong iron chelator[60,61]. Age- or Alzheimer's-related atrophy of neuromelanin-containing cells would attenuate MTsat and R2*. This atrophy may lead to an overall reduction in neurite density across the brain as well as localized changes in neurite dispersion as IdC axonal arborizations atrophy. This may even be detectable earlier than microstructural changes within IdC due to Wallerian degeneration whereby axonal processes degrade before cell bodies[62].

Although MTsat may still represent cell density in DR and NbM, these regions do not contain neuromelanin. R2*, although highly

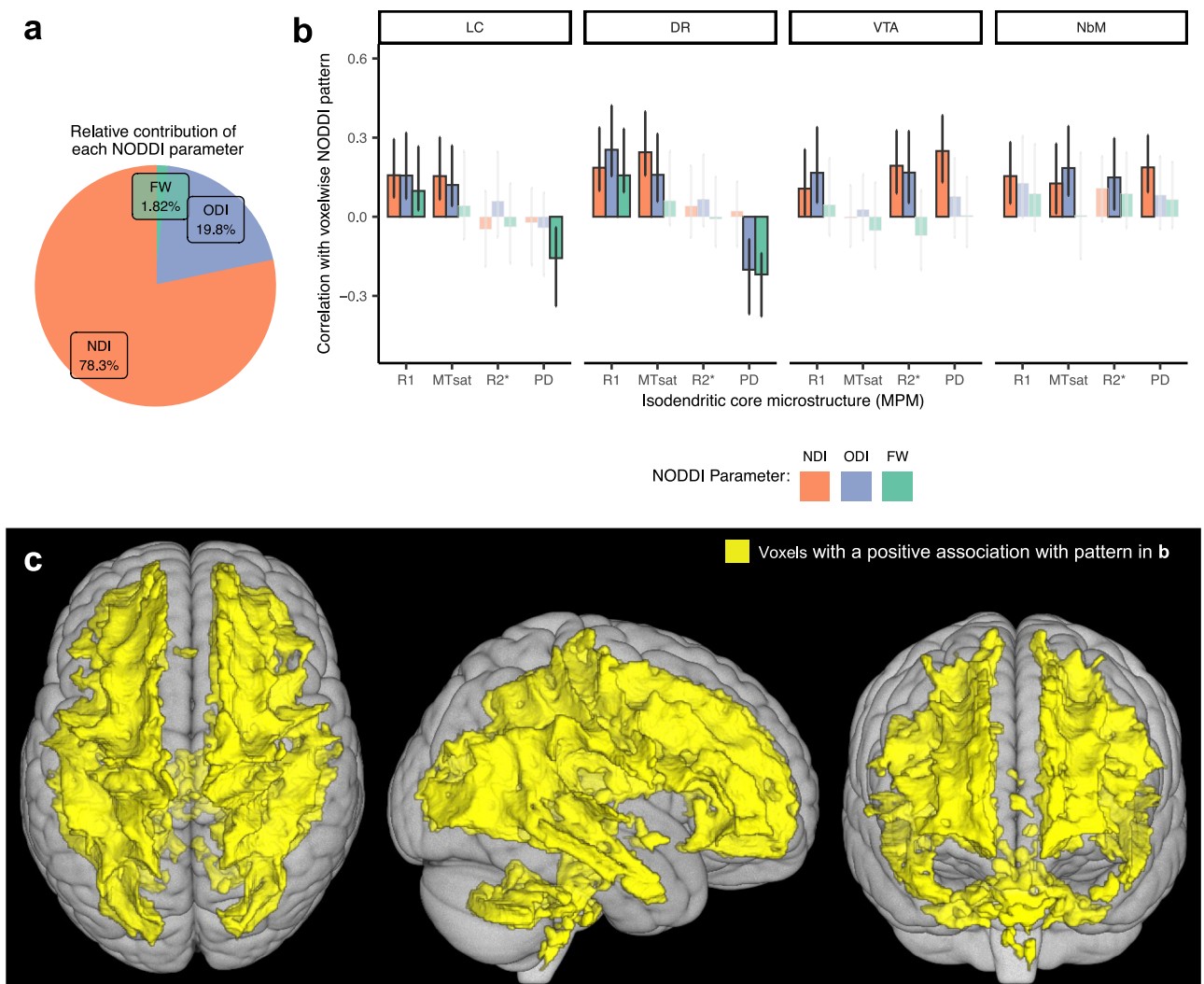

**Fig. 3 | LV1 correlations between IdC MPM measures and NODDI in white matter. a** The relative contributions of each NODDI metric to the overall pattern are calculated as a sum of the design scores and expressed as a percentage. NDI is the primary driver of this pattern. **b** The strength and direction of the relationship that each measure of isodendritic core microstructure (R1, MTsat, R2*, PD) has with the voxels highlighted in (**c**) for each measure of white matter microstructure (NDI, ODI or FW). Error bars show 95% confidence intervals from 1000 bootstrapped samples. The midpoint of each bar represents the correlation using the full dataset. Non-significant correlations (confidence intervals overlapping with zero) are faded out. **c** The yellow color of the voxels indicates a positive relationship with the pattern shown in the top panel. Only voxels with a bootstrap ratio of >|2| are colored. Here, we see strong positive associations between R1 in all isodendritic core nuclei and NDI (as well as ODI) across most of the white matter. LC locus coeruleus, DR dorsal raphe, VTA ventral tegmental area, NbM nucleus basalis of Meynert, MTsat magnetization transfer saturation, PD proton density, NDI neurite density index, ODI orientation dispersion index, FW free water fraction. Source data are provided as a Source Data file.

sensitive to iron content, is also sensitive to diamagnetic materials, e.g., myelin[45]. A positive covariance between R1, MTsat and R2* is therefore suggestive of variation in myelin content. Myelination within IdC nuclei may be the driving factor behind covariation with whole-brain white matter. Demyelination may represent an early marker of vulnerability within these regions. Associations between IdC and white matter microstructure are not necessarily causally linked. A common factor (e.g. subtle global Alzheimer's-related pathology) may drive IdC-white matter covariance. Further studies are required to determine the exact nature of these associations and the presymptomatic pathological staging of AD.

### APOE4-specific widespread variation in white matter neurite complexity

R1, MTsat, and R2* across all four IdC nuclei were positively associated with NDI in widespread white matter regions, which covaried with ODI. In LC and DR, this association was strongest with R1 and MTsat, indicating driving factors such as cell density, neuromelanin or lipid content within these nuclei. In the VTA the association was driven by R2* and PD, indicative of variation in iron and water content, respectively. This pattern was largely specific to *APOE4* carriers.

Decreased global NDI as a function of white matter hyperintensities has been previously reported in a cohort of older adults[63]. It is possible that our pattern is driven by similar factors, e.g., neuroinflammation. Indeed, the specificity of this pattern to the *APOE4* genotype may be due to the role of APOE in blood-brain-barrier maintenance, vascular health, as well as inflammation regulation[64]. In line with our findings, a more widespread pattern of NDI reductions in people with young-onset AD *APOE4* carriers compared to non-carriers has been previously reported[65].

Positive covariance of NDI and ODI is indicative of variation in neurite complexity. Neurite dystrophy, as seen in AD[66], can reduce the number of neurite processes (decreased NDI) while also reducing branching and increasing fiber directionality (decreased ODI). Variation in neurite complexity is not necessarily specific to AD[67] but is

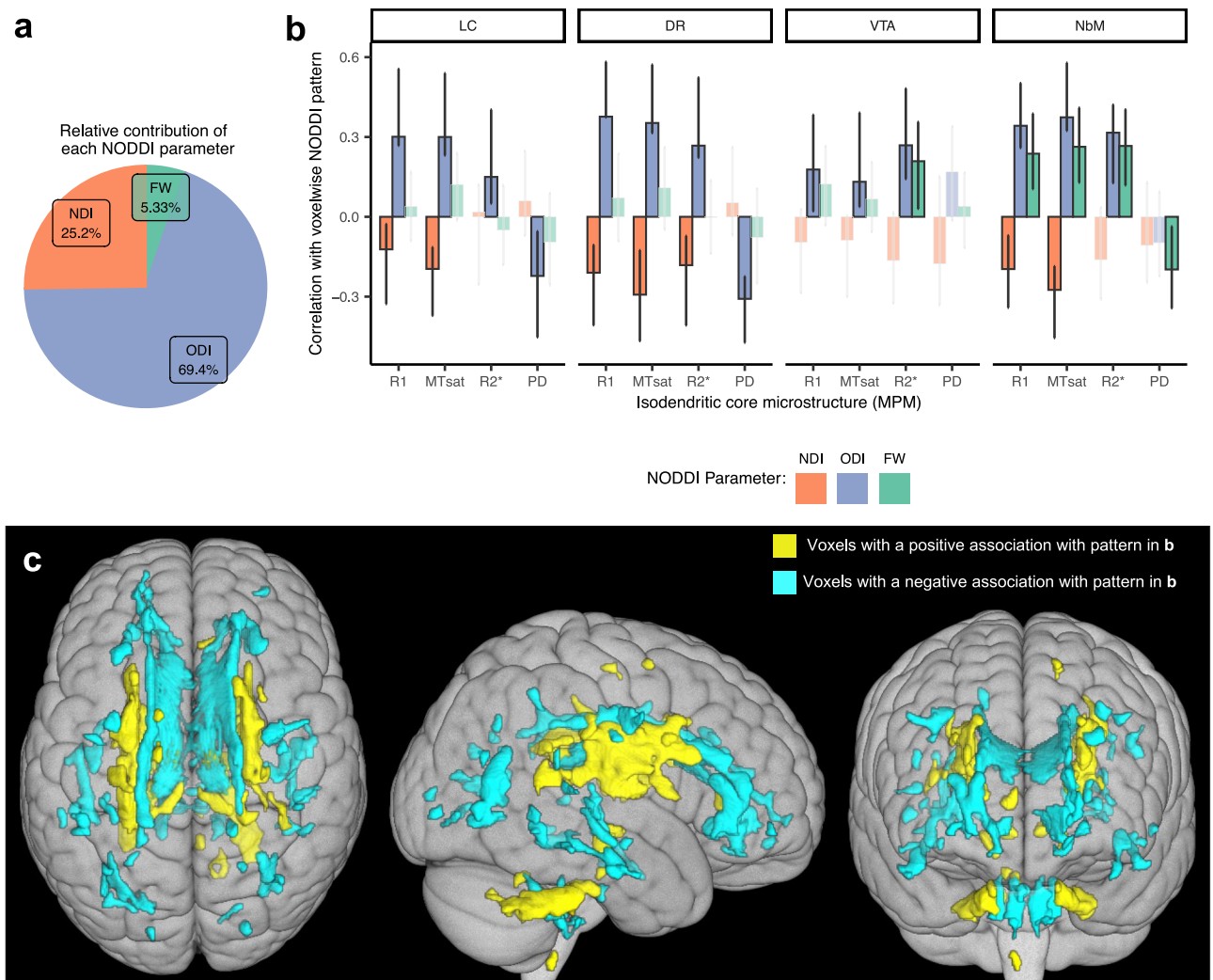

**Fig. 4 | LV2 correlations between IdC MPM measures and NODDI in white matter. a** The relative contributions of each NODDI metric to the overall pattern are calculated as a sum of the design scores and expressed as a percentage. Covariance of ODI with IdC integrity is the primary driver of this pattern. **b** The strength and direction of the relationship that each measure of isodendritic core microstructure (R1, MTsat, R2*, PD) has with the voxels highlighted in (**c**) for each measure of white matter microstructure (NDI, ODI or FW). Error bars show 95% confidence intervals from 1000 bootstrapped samples. The midpoint of each bar represents the correlation using the full dataset. Non-significant correlations (confidence intervals overlapping with zero) are faded out. **c** The yellow voxels indicate a positive relationship with the pattern shown in (**b**). The blue voxels indicate a negative relationship. Only voxels with a bootstrap ratio of >|2| are colored. These plots can be interpreted as follows: Greater R1 in LC is associated with lower NDI in yellow regions (brainstem efferent tracts) and greater NDI in blue regions (limbic tracts), as the correlation value in (**b**) is negative. Greater R1 in LC is also associated with greater ODI in yellow regions and lower ODI in blue regions, as the correlation value in (**b**) is positive. R1 in LC is not associated with FW in this pattern, as the correlation value confidence intervals in (**b**) overlap with zero. Overall, in this pattern, we see positive associations between R1, MTsat and R2* across all isodendritic core nuclei and ODI in yellow regions (brainstem efferent tracts) and negative associations with ODI in blue regions (limbic tracts). LC locus coeruleus, DR dorsal raphe, VTA ventral tegmental area, NbM nucleus basalis of Meynert, MTsat magnetization transfer saturation, PD proton density, NDI neurite density index, ODI orientation dispersion index, FW free water fraction. Source data are provided as a Source Data file.

associated with many other disorders and risk factors, e.g., cardiovascular health[64] and cognitive reserve[68]. Indeed, we see an association between the expression of this pattern and years of education. Overall, we are interpreting this pattern as a feature of global brain health.

A notable exception from this global pattern is voxels within midline regions such as corpus callosum, indicating a different relationship with the isodendritic core than the rest of cerebral white matter. The tightly bundled, highly directional microstructure of corpus callosum may allow for less variation in neurite density (and therefore is not described by pattern 1) while allowing for variation in orientation dispersion (and is therefore described by pattern 2). Further exploration and validation of this relationship would be an interesting topic for future study.

### AD-specific variation in white matter integrity

We observed a second pattern that specifically related IdC integrity to Alzheimer's-vulnerable white matter regions. This pattern was characterized by a positive covariance between R1, MTsat, and R2* in all IdC nuclei, positively related to ODI within white matter regions with strong direct brainstem connections[69] (internal capsule, corona radiata), and negatively related with limbic tracts (cingulum bundle). Specific to LC and DR, we found a negative association of PD with the other measures of IdC integrity. ODI negatively covaried with NDI consistent with the effects of neurodegeneration[51]. This pattern was more pronounced in *APOE4* carriers.

We suggest that this pattern reflects presymptomatic Alzheimer's-specific pathology. The highlighted white matter tracts all show microstructural changes in the early stages of AD[55–57]. Furthermore,

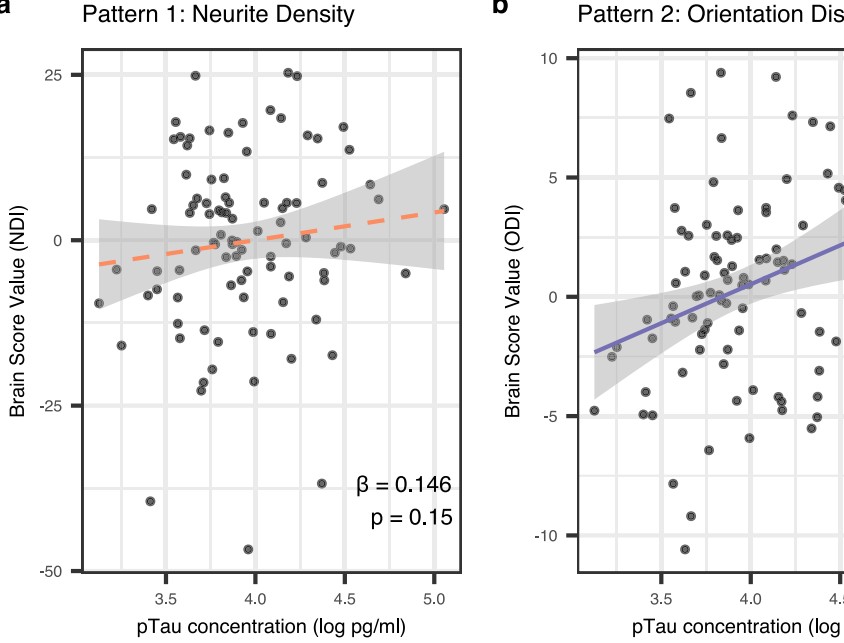

**Fig. 5 | Scatterplots between CSF pTau181 concentration and PLS brain scores for (a) NDI in pattern 1 and (b) ODI in pattern 2.** NDI neurite density index (orange), ODI Orientation dispersion index (purple). Higher brain scores indicate stronger expression of the covariance pattern. Standardized slope coefficients (β) and *p*-values are shown on each respective plot. Regression lines are shown with standard error shown in gray. All statistical tests shown were two-tailed. Source data are provided as a Source Data file.

people with greater CSF pTau181 concentration expressed this pattern more strongly. Critically, this pattern is distinct from the effects of normal ageing. The relevance of this pattern to AD is also supported by its stronger expression in *APOE4* carriers.

The same white matter regions (limbic and brainstem efferent tracts) are affected in both *APOE4* carriers and non-carriers with young-onset AD[65]. Specifically, reductions in ODI in the corona radiata and internal capsule compared to controls in both genotypes were observed, but with an increase in FW in the corpus callosum specific to the *APOE4−* group. In line with this, our *APOE4* groupwise analysis reveals a third pattern, specific for the *APOE4−* group, involving similar white matter regions to those in the second (*APOE4+*) pattern but with stronger associations with FW. Carriers versus non-carriers, therefore, show overlapping but distinct patterns of white matter degeneration with primary variation in ODI or FW, respectively, that is related to IdC integrity.

This *APOE4−* pattern also highlighted microstructural covariance in the IdC that was restricted to LC and DR. In contrast, the *APOE4+* pattern (LV2) involved all four IdC nuclei, suggesting *APOE4−* individuals may show relatively restricted microstructural variation within IdC compared to *APOE4+* carriers. LC and DR also interact with the *APOE4* genotype (LV4): integrity in LC and DR is associated with a distributed pattern of voxels in limbic tracts and peripheral white matter in opposite directions for each genotype. IdC nuclei may be differentially vulnerable in Alzheimer's pathology depending on *APOE4* genotype, with VTA and NbM expressing greater *APOE4*-related risk than LC and DR. Further study of these nuclei and neuromodulatory systems, particularly dopaminergic and cholinergic systems, may be critical to understanding the mechanisms behind *APOE4*-related Alzheimer's risk.

A mechanism for this *APOE4* specificity may lie in the role of *APOE4* in myelin dysregulation[53]. IdC axons are poorly myelinated in higher mammals (e.g. primates)[70,71]. Given that *APOE4+* genotypes have impaired myelination compared to *APOE4−* genotypes[53], less 'redundancy' in their myelination may cause the IdC neurons to be particularly sensitive to the dysregulation of myelin that arises from *APOE4*. *APOE4* may also mediate disease through altered inflammatory states

and vascular dysfunction[72]. Damaged cerebral vasculature or increased presence of inflammatory mediators may disproportionately affect the isodendritic core, given the greater metabolic demands of the long and highly arborized neurons[5–7]. Future studies should address the *APOE4* genotype's impact on myelination, inflammation, and vascular dysfunction in relation to the vulnerability of IdC projections in humans.

**Complex changes in orientation dispersion with early pathology.** A critical feature of the second (Alzheimer's-related) pattern was the dissociation between limbic and brainstem-efferent tracts. In contrast to NDI, ODI does not have a clearly agreed-upon 'direction of change' due to neurodegeneration. Greater ODI in the cingulum in a group with MCI compared to healthy controls has been reported[73]. Greater ODI in corpus callosum with older age has been shown alongside a lower ODI in corona radiata with greater volume of white matter hyperintensities, a measure of cerebrovascular pathology[63]. In contrast, spatially localized reductions of ODI in MCI and AD compared to controls have been reported elsewhere[74].

The contrasting effects of ODI within limbic versus brainstem-efferent tracts are likely due to the effects of subtle degeneration on tracts with different microstructural properties. For example, in a bundle where fibers are largely coherently aligned (e.g., cingulum), degeneration of the main fiber population would cause greater ODI as crossing fibers gain relatively higher importance. In contrast, degeneration of a highly branched tract (e.g., corona radiata) would cause a lower measure of ODI as fanning is reduced.

Non-monotonic trajectories of diffusion metrics across different stages of Alzheimer's pathology may further complicate the picture[63,75–78]. ODI has been shown to initially decrease before later increasing in concert with macroscopic neurodegeneration. The earliest changes may represent compensation mechanisms or complexly developing neuroinflammatory processes.

**Limitations**

The primary limitation of this study is the small size of the IdC nuclei relative to the image resolution (1 mm³). Acquiring higher-resolution

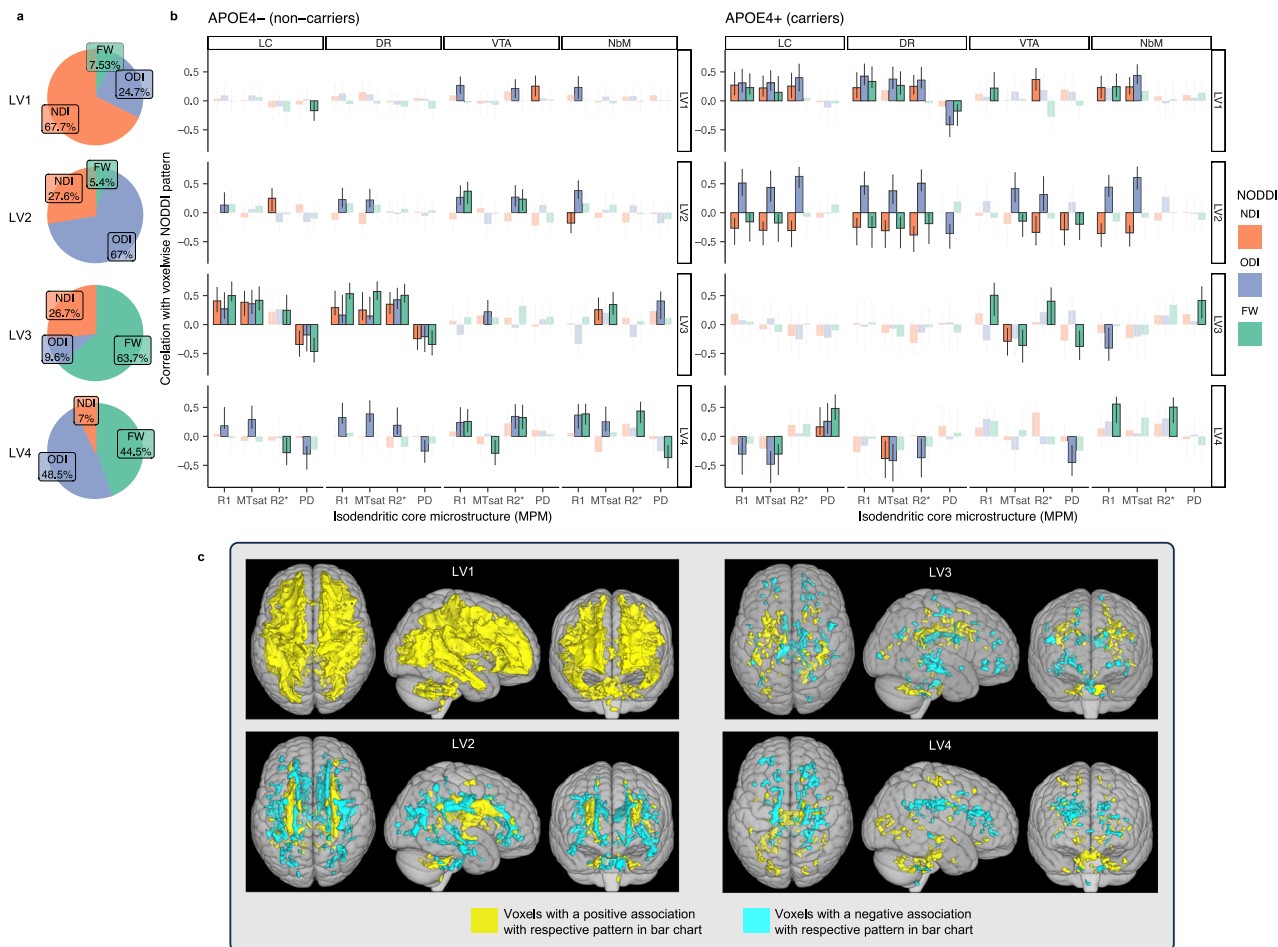

**Fig. 6 | PLS results for *APOE4* group analysis.** Pie charts (**a**) show the relative contributions of each NODDI metric to each LV, calculated as a sum of the design scores and expressed as a percentage. The bar charts (**b**) show the strength and direction of the correlation between each measure of isodendritic core microstructure (R1, MTsat, R2*, PD) with the voxels highlighted in panel **c** for each measure of white matter microstructure (NDI, ODI, FW). These data are shown separately for *APOE4-* (left **b**) and *APOE4+* (right **b**) groups. The yellow voxels highlighted in **c** indicate a positive relationship with the pattern shown in panel **b**. The blue voxels indicate a negative relationship. Only voxels in **c** with a bootstrap ratio of >|2| are colored. Non-significant correlations in **b** are faded out. Error bars show 95% confidence intervals from 1000 bootstrapped samples. The midpoint of each bar represents the correlation using the full dataset. Here, we see the overall specificity of the first two patterns to the *APOE4* positive group and the specificity of the third pattern to the *APOE4* negative group. LC locus coeruleus, DR dorsal raphe, VTA ventral tegmental area, NbM nucleus basalis of Meynert, MTsat magnetization transfer saturation, PD proton density, NDI neurite density index, ODI orientation dispersion index, FW free water fraction, LV latent variable. Source data are provided as a Source Data file.

images was limited by signal-to-noise and scan time restraints. Some partial voluming from neighboring regions is unavoidable, however we chose to threshold probabilistic masks to minimize this as much as possible while still leaving masks with enough voxels from which to calculate a reliable measure of signal intensity. We also controlled for a non-IdC ROI in our analyses to increase the specificity of our conclusions to the IdC nuclei. Finally, we upsampled our images to 0.5 mm to make use of the higher-resolution atlases provided. While new structural information cannot be created by upsampling resolution, it can increase the precision of ROI boundaries. We acknowledge the benefits of higher-resolution imaging for the nuclei of the isodendritic core and consider it a valuable direction for future investigations to better understand these regions and their relationship to aging and neurodegenerative disease.

A related limitation is the reliance on standard-space atlases instead of individualized masks, as boundaries for these ROIs are mostly invisible on the scan types used. These regions are relatively evolutionary conserved, and structure is likely to be similar between individuals after accounting for gross variation in brainstem morphology (as our registration pipeline does). Our sample size is sufficient to average out small idiosyncrasies in individual IdC

morphology not accounted for by our registration pipeline. Future studies could improve precision of masking using PET tracers for individual neuromodulator markers that highlight each nucleus, however this is prohibitively expensive and cumbersome for most studies.

To ensure we were only exploring patterns within white matter and not neighboring CSF or gray matter regions, we strictly thresholded our white matter mask to include only voxels identified as white matter for at least 95% of subjects. This criterion prioritized voxel specificity at the cost of sensitivity and, therefore, excluded very peripheral white matter areas as well as thinner isolated structures. Unfortunately, this meant excluding certain regions of potential interest, such as the fornix, which has been previously implicated in aging and Alzheimer's disease. Future studies specifically examining fornix are of great interest and relevance to this field.

Finally, as with any correlational study, the presence or direction of causality cannot be determined from these findings. Longitudinal and intervention studies should explore these regions and their relationship with aging and neurodegenerative disease with the specific aim to determine causative mechanisms.

## Conclusions

Our study highlights patterns of microstructural covariance between neuromodulatory nuclei of the IdC and whole-brain white matter. IdC microstructure covaried with two spatial patterns across the white matter, which, respectively, represent (1) general brain health and (2) early AD pathology. The latter pattern was associated with CSF pTau181 concentration, and both were more pronounced in *APOE4* carriers. *APOE4* non-carriers expressed a restricted pattern of covariance between IdC microstructure and limbic white matter tracts.

Structural relationships between the IdC nuclei and the white matter through which they project may be some of the earliest detectable brain changes in AD. Here we have capitalized on advances in quantitative MRI to make inferences about such relationships with greater biological interpretability than traditional methods. Our conclusions support the characterization of myelin within these nuclei as a target for disease tracking. We also highlight that dopaminergic and cholinergic nuclei are specifically vulnerable in *APOE4* carriers compared to non-carriers. Improved understanding of early neuropathological changes in these systems holds the potential to extend the window for early surveillance, diagnosis and disease staging, opening new avenues for prevention, treatments and disease-modifying therapies.

Validation of these results with submillimeter resolution images (afforded through ultra-high field strength MRI) and individualized masking of IdC nuclei is of great value to the clinical translation of this work. Furthermore, it would be valuable to examine whether certain characteristics of white matter integrity may be sensitive and specific enough to IdC pathology that white matter alone may be the target of clinical focus. This may facilitate early signs of AD detection using clinically viable MRI field strengths.

## Methods

### Participants

141 participants with a first-degree familial history of AD were included in the PResymptomatic EVAluation of Experimental or Novel Treatments for AD (PREVENT-AD) cohort[79]. All participants gave informed written consent before participating in the study. The procedures of the PREVENT-AD study were approved by the McGill institutional review board and/or the Comité d'éthique de la recherche du CIUSSS de l'ouest de l'île de Montréal. The study was performed in accordance with the ethical standards laid down in the 1964 Declaration of Helsinki. All participants included in this study were recruited as part of PREVENT-AD recruitment protocols in 2011–2017. These included extensive advertising through multiple available media and community-access platforms across Montreal and the surrounding areas in Quebec, Canada. More information about the PREVENT-AD program can be found in Supplemental Materials.

Eight participants were excluded due to poor imaging data or registrations. Of the remaining participants, 133 had MPM and diffusion MRI data (details in the following section: 'Image protocol and processing'). Participant age averaged $67.9 \pm 5.3$ years, including 95 females, with an average of $15.4 \pm 3.5$ years of education.

### Imaging protocol and processing

MRI scans were acquired on a 3 T Siemens PrismaFit at the Douglas Research Centre, including a T1-weighted anatomical scan (Magnetization-Prepared Rapid Acquisition Gradient Echo (MPRAGE), 1 mm isotropic resolution, TR/TE/TI = 2300/2.96/900 ms, FA = 9°, TA = 5:30), MPM and multi-shell diffusion-weighted imaging sequences (described below). All sequences had whole-brain coverage, including the brainstem.

**Multi-parametric mapping.** In this study, we used MPM to measure R1 relaxation rate (1/T1), magnetization transfer saturation (MTsat), R2* relaxation rate and proton density (PD), each of which is differentially sensitive to different elements of tissue microstructure such as lipid, iron and water content. The MPM sequence was developed and provided by the McConnell Brain Imaging Centre of The Neuro.

**MPM acquisition parameters.** For MPM, three multi-echo gradient echo sequences were acquired (1 mm isotropic resolution, TA = 17:30) with predominant weighting for: T1 (TR = 18 ms, 6 echoes, TE = 2.16–14.81 ms, FA 20°), MT (TR = 27 ms, 6 echoes, TE = 2.04–14.89 ms, echo-spacing = 2.57 ms, FA 6°) or PD (TR = 27 ms, 8 echoes, TE = 2.04–22.20 ms, echo-spacing = 2.57 ms, FA 6°).

B1+ transmit field maps were acquired using two spin-echo echo-planar sequences with different flip angles (60°, 120°) and otherwise identical parameters ($2 \times 2 \times 4$ mm resolution, TR/TE = 4010/46 ms, TA = 1:08).

**MPM processing.** B1+ maps were created in qMRLab[80] using the double-angle method[81] with a sixth-order polynomial smoothing kernel. All further processing of MPM maps was performed using the hMRI toolbox[82] (v 0.4.0). Maps for MTsat, R1, R2* and PD were corrected for B1− receive field inhomogeneities, calculated from two PD-weighted turbo-flash sequences acquired using either the body coil or 32-channel head coil, with otherwise identical acquisition parameters (2 mm isotropic resolution, TR/TE = 344/1.55 ms, FA = 3°, TA = 0:35). Default hMRI toolbox settings were used, with the exception that R2* maps were estimated using log-linear weighted least-squares ('WLS1')[83]. Parameter maps were denoised using ANTS DenoiseImage[84].

*Regions of interest*: Previously validated atlases were used to define LC[85] (thresholded at 10%), VTA[86] (thresholded at 25%) and NbM[87,88] (thresholded at 60%) (Supplementary Fig. 1). Thresholds were chosen to limit overlap into neighboring structures, increasing specificity of the extracted values to the actual region. Greater threshold limits reduce the number of included voxels that do not belong to the region of interest (increasing specificity), while lower thresholds increase the certainty that the entire region is covered at the expense of some surrounding regions also being included (increasing sensitivity). Given that qMRI measures can differ greatly in neighboring structures (especially CSF), we chose to increase specificity and the slight expense of sensitivity. Atlases not provided in MNI ICBM152 asymmetrical 2009b 0.5 mm standard space (VTA and NbM) were resampled from ICBM152 asymmetrical 2009c 1 mm standard space[89] using ANTS[90].

The DR mask was manually defined. Although publicly available masks of DR do exist[91–93], we observed large discrepancies between their spatial profiles, with little overlap between them in some cases. For example, the mask of Bianciardi et al.[92] (thresholded at 50%) extends considerably more rostrally, up to the level of the cerebral aqueduct and superior colliculus, than those of Levinson et al.[91] (thresholded at 50%) and Edlow et al.[93] (Supplementary Fig. 2). These discrepancies may be due to biases toward different DR subnuclei, resulting from methodological differences in how the region is localized (explained further in supplemental information). Given these discrepancies, we decided to leverage the BigBrain histological atlas[94]. This atlas allows for visualization of the serotonergic cell bodies of the DR, and to directly translate their location to MNI coordinates thanks to a previously published precise registration pipeline[95]. Combining this information with histological descriptions of DR[96,97] and the Allen Human Brain Atlas[98], a spherical ROI (3 mm radius) was drawn around MNI coordinates x: 0, y: −30, z: −13. Any edge voxels close to the cerebral aqueduct of periaqueductal gray were trimmed, leaving a roughly spherical ROI of volume ~32 mm³. This position was chosen to align with the supratrochlear subnucleus of DR, where neurofibrillary tangles have been identified in asymptomatic stages of Alzheimer's disease[14]. Our mask is available online (see ref. 99).

We also defined a bilateral spherical pontine ROI that did not overlap with any part of the IdC as a methodological control region (Supplementary Fig. 3).

While this study includes the examination of VTA, we did not examine the substantia nigra, another monoaminergic nucleus often discussed in the context of Alzheimer's disease. Substantia nigra rarely exhibits significant atrophy compared to the rest of the isodendritic core, and there is less evidence for early tauopathy and degeneration in substantia nigra than in VTA (see ref. 3 for a review). In contrast, dopaminergic neuronal loss has been localized specifically to VTA (while absent in substantia nigra) in a mouse model of Alzheimer's disease[100]. Furthermore, in vivo, evidence in humans suggests that above and beyond any association with substantia nigra, VTA exhibits a lower T1-weighted signal, indicating microstructural changes, in individuals who experience more cognitive decline compared to cognitively stable individuals[21]. Finally, in the earliest stages of Alzheimer's disease, there is evidence to suggest that substantia nigra is more susceptible to alpha-synuclein deposition than neurofibrillary tangles[101] and, therefore, may represent different disease mechanisms or co-pathology. In support of this, substantia nigral pathology is more closely related to the presence of extrapyramidal motor symptoms such as dyskinesia and Parkinsonism, which tend to become more prevalent in later stages of disease[101]. In contrast, VTA atrophy is more closely associated with cognition deficits[23]. For these reasons, we focused on VTA over substantia nigra. However, the individual roles of both nuclei in the early and late stages of Alzheimer's disease and their relation to comorbidities require considerably more work to untangle.

*Normalization of MPM maps to MNI space*: The following registration protocol was implemented to ensure high-quality alignment between native space scans and standard space (ICBM 2009b). Registrations were performed using ANTS (antsRegistrationSyNQuick.sh)[90]. For each subject, the denoised R1 map was registered (affine) to the denoised MPRAGE. Then, each subject's MPRAGE was normalized to the standard-space T1w template using a previously described two-step process[102,103]. First, rigid and then affine registrations were calculated with the denoised MPRAGE as the moving image and the MNI template as the fixed image. This created an approximate alignment between the images. Second, this approximately aligned MPRAGE was registered to the template image again, this time using subsequent rigid, affine, then non-linear (SyN) warps, the latter stage of which used a manually defined registration mask of the brainstem, midbrain and basal forebrain area (Supplementary Fig. 2). This ensured good quality alignment within this region (confirmed by visual inspection of all cases). As a final step, warps were inverted and concatenated to transform MNI-space ROIs into each subject's native MPM-space (with nearest neighbor interpolation). Partial volume effects with neighboring regions may still slightly influence measures of integrity within each ROI.

**Neurite orientation dispersion and density imaging.** White matter microstructure was assessed using NODDI[52]. NODDI improves the specificity of the derived measures by leveraging multi-shell diffusion encoding and a multi-compartment biophysical model of brain tissue.

**Diffusion imaging acquisition parameters.** Diffusion-weighted imaging was performed with a 2 mm isotropic spin-echo echo-planar imaging sequence with 109 measurements (isotropically spaced around a sphere) across three shells with the following $b$-values: 7 $b = 300$, 29, $b = 1000$ and 64, $b = 2000$ s/mm$^2$, as well as 9 $b = 0$ images (TR/TE = 3000/66 ms, posterior-anterior phase encoding, TA = 5:49). Five additional b = 0 images were acquired with reversed phase encoding direction and otherwise identical sequence parameters for subsequent distortion correction.

**NODDI processing.** Diffusion images were preprocessed using the MRtrix3 toolbox[104]. Images were denoised (*dwidenoise*)[105–107], corrected for eddy, susceptibility and motion artifacts (*dwifslpreproc*)[108–110] and upsampled to 1 mm isotropic resolution (*mrgrid*). Upsampled images were brain-extracted (*bet2*)[111], corrected for bias-field artifact using the

ANTs algorithm (N4)[112] of the *dwibiascorrect* MRtrix function. Bias field corrected diffusion-weighted data were fitted to the NODDI model using the Python implementation of Accelerated Microstructure Imaging via Convex Optimization (AMICO)[52,113]. The response functions were computed for all compartments, and fitting was then performed on the bias-corrected diffusion-weighted volumes within a brain mask. The resulting parameters obtained are NDI, ODI and FW.

*Region of interest:* We sought to examine NODDI parameters throughout the entire brain's white matter. A probabilistic mask of each individual's normal-appearing white matter was made using the *Sttgen* function of MRtrix3[114].

*Normalization of NODDI map to MNI space*: Upsampled NODDI maps and probabilistic white-matter masks were co-registered with MPM maps using FSL '*flirt*' (rigid body registration, using $b = 0$ volumes as the moving image and PD maps as a reference image due to similar contrast). MTsat maps were segmented into gray matter, and white matter tissue compartments, and then deformation fields were estimated between native space and MNI (2009c) space using the '*Shoot*' toolbox in SPM12[115]. The resulting deformation fields were then applied to warp the MPM-space NODDI maps and white-matter masks, thereby bringing them into the MNI template space. White matter masks warped to MNI space were averaged across individuals and thresholded at 0.95. Prior to the main analysis, we 'regressed out' the age and microstructure of a non-isodendritic pontine control region (measured by both NODDI and MPM parameters) from the NODDI maps using multiple regression, implemented in SPM12. The pontine ROI control was included to ensure any effects observed were specific to relationships between white matter and IdC nuclei, as opposed to any other given brain region, as this ROI is not known to be affected early in AD. The pontine ROI is shown in Supplementary Fig. 3. Images were then smoothed with a 1 mm full-width half-maximum Gaussian kernel. The resulting residual images were used as inputs for the partial least squares (PLS) analysis (see below).

## APOE4 genotyping

Methods for participant genotyping have been described previously[79]. Briefly, DNA was isolated from 200 µl whole blood using a standard QIASymphony apparatus and the DNA Blood Mini QIA Kit (Qiagen, Valencia, CA, USA), run according to the manufacturer's instructions. Allelic variants, including *APOE* rs429358 and rs7412, were determined using pyrosequencing (PyroMark24 or PyroMark96) or DNA microarray (Illumina). Given the small number of 4/4 subtypes, participants were classed as having no e4 alleles (*APOE4*−) or has at least one e4 allele (*APOE4*+). One subject did not have *APOE* genotype data available, leaving 132 participants in this analysis. Of these, 50 people had at least one *APOE4* allele and 82 did not. This is a greater proportion of carriers than would be found in the general population, explained by the fact that these individuals have been specifically recruited to have familial risk (first-degree family history) for Alzheimer's disease.

## Cerebral spinal fluid (CSF) biomarkers

Methods for extraction of CSF from participants have been described previously[79]. Briefly, up to 30 ml of CSF was withdrawn using a lumbar puncture between L3-L4 or L4-L5 using an atraumatic Sprotte 24 ga. spinal needle. Samples centrifuged at room temperature for 10 minutes at 2000g then stored at −80 °C. Amyloid-beta 1-42 (Aß42) and phosphorylated tau at threonine 181 (pTau181) concentrations were determined by enzyme-linked immunosorbent assay using Innotest technology (Fujirebio). CSF data were available for 93 subjects (with comparable demographics to those without CSF data, shown in Supplementary Table 1). CSF was collected prior to MRI data, with a median delay of 768 ± 375 days. Due to this delay, the time between CSF and MRI data collection was explored as an interaction term in statistical models where appropriate, but in no case did it significantly

improve model fit (measured by Akaike Information Criterion), so it was excluded from all models.

## Statistical analysis

To explore the relationship of IdC with demographic variables and CSF biomarkers we ran partial correlations, correcting for demographic covariates (age, sex, years of education).

We use partial least squares (PLS) analysis to assess the multi-variate spatial patterns of covariance between four MPM-derived measures of microstructural integrity (R1, MTsat, R2*, PD) in the four IdC nuclei (LC, DR, VTA, NbM) and three NODDI measures of white matter microstructure (NDI, ODI, FW). Each NODDI-parameter map had age and MPM-parameters of the pontine ROI regressed-out prior to analysis and was entered as a separate condition in a single PLS analysis. We did not examine MPM-derived measures across the whole-brain white matter in order to focus our findings within the context of the NODDI model. This approach enabled us to more specifically examine relationships between the IdC and white matter microstructure.

The PLS was run with 1000 permutations to determine the significance of each LV and 1000 bootstraps to determine overall reliability of each voxel's contribution to each LV by calculating the standard error of each voxel's salience value. Only significant ($p < 0.05$) LVs and voxels with bootstrap ratios >|2| (calculated as the ratio of each voxel's salience to its standard error) were interpreted. Bootstrap ratios are equivalent to a $z$ statistic, and values > |2| are roughly equivalent to a p-value < 0.05. Clusters smaller than 32 voxels (default of MRIcroGL viewing software) were also excluded. Individual effects of IdC MPM parameters were deemed significant if 95% bootstrap confidence intervals on correlation coefficients did not overlap with zero. PLS facilitates the examination of covariance between numerous variables in a single model and a single analytic step without the need for multiple-comparisons correction. In order to visualize which NODDI parameter contributed most to each LV, we summed design scores across all IdC-MPM parameters for each NODDI condition and display them as a pie chart alongside correlation summary plots. PLS analyses were performed in MATLAB 2022a (Mathworks Inc.)[54].

For each of the three NODDI PLS conditions (NDI, ODI, FW), we extracted brain scores expressing significant LVs. We used linear regression models to test whether these brain scores were related to Alzheimer's pathology. For each model, the brain score served as the dependent variable, and CSF biomarker concentration (either pTau181 or Aβ42) served as the independent predictor variables, with sex and years of education as covariates (age was already residualized out of the NODDI maps, so was not included here as well). $p$-values from these two tests were Bonferroni corrected for multiple comparisons (shown as $p_{adj}$). All statistical tests were two-tailed ($\alpha = 0.05$). These analyses were performed in R v4.2.1. We also ran an additional PLS correlation analysis between CSF pTau and NODDI maps to explore direct relationships between white matter microstructure and Alzheimer's disease pathology (see Supplementary Materials).

## Reporting summary

Further information on research design is available in the Nature Portfolio Reporting Summary linked to this article.

## Data availability

This study used data from the PResymptomatic EValuation of Experimental or Novel Treatments for AD (PREVENT-AD) study[79]. Data availability for PREVENT-AD is governed by the Open Access protocols. Please refer to https://douglas.research.mcgill.ca/prevent-alzheimer-program/ for more information about PREVENT-AD. Source data are provided in this paper. Source data are provided with this paper.

## Code availability

This study utilized a number of open-access resources. Multi-parametric maps were processed using qMRLab[80] (https://github.com/qMRLab/qMRLab) and hMRI toolbox[82] (v 0.4.0) (https://hmri-group.github.io/hMRI-toolbox/) in MATLAB. Diffusion images were pre-processed using the MRtrix3 toolbox: https://www.mrtrix.org/[104]. We ran PLS using an openly available MATLAB toolbox available to download here: https://github.com/McIntosh-Lab/PLS/[54]. Partial correlations were performed using the openly available *ppcor* R package (v1.1) in R: https://CRAN.R-project.org/package=ppcor and visualized using the *corrplot* R package (v0.92). All other data presented in the figures were organized and plotted using functions from the *tidyverse* R package (v2.0.0): https://github.com/tidyverse. The voxelwise salience maps were displayed using MRIcroGL: https://www.nitrc.org/projects/mricrogl.

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

## Acknowledgements

We would like to thank the PREVENT-AD research team and study participants for their time and dedication in collecting the data used in this study. A full list of PREVENT-AD authors and collaborators can be found at https://preventad.loris.ca/acknowledgements/acknowledgements.php?date=[2023-07-01] Full acknowledgments can be found in Supplemental Materials. We would also like to thank the Laboratory of Brain and Cognition at the Montreal Neurological Institute for productive discussions and helpful feedback. This project was made possible by financial support from the Alzheimer's Association (AARG-22-927100) and the National Institutes of Health (NIA R01 AG068563). R.N.S. is a Research Scholar supported by the Fonds de la Recherche du Quebec – Santé (FRQS). A.W. is a postdoctoral fellow also supported by FRQS.

## Author contributions

A.W.: Conceptualization, methodology, software, formal analysis, data curation, writing—original draft, writing—review & editing, visualization, funding acquisition. S.A.T.: Software, formal analysis, writing—review & editing. C.L.T.: Conceptualization, methodology, resources, supervision, funding acquisition, writing—review & editing. I.R.L.: Methodology, investigation, writing—review & editing. C.G.: Supervision, writing—review & editing. G.B.: Data curation, writing—review & editing. C.H.: Data curation, writing—review & editing, P.H.: Data curation, writing—review & editing. J.T.-M.: Investigation, writing—review & editing, P.R.-N.: Investigation, resources, funding acquisition, J.P.: Investigation, resources, funding acquisition, S.V.: Investigation, resources, funding acquisition, writing—review & editing, T.W.S.: Conceptualization, writing—review & editing, G.R.T.: Conceptualization, supervision, writing— review & editing. R.N.S.: Conceptualization, resources, writing—original draft, writing—review & editing, supervision, project administration, funding acquisition.

## Competing interests

The authors declare no competing interests.

## Additional information

## PREVENT-AD Research Group

**Christine L. Tardif**[1,5,6]**, Ilana R. Leppert**[1,5]**, Jennifer Tremblay-Mercier**[8]**, Pedro Rosa-Neto**[1,5,8]**, Judes Poirier**[8,9]**, Sylvia Villeneuve**[5,8,9] **& R. Nathan Spreng**[1,5,8,9] ✉

A full list of members and their affiliations appears in the Supplementary Information.

