## [Peer Review File · Nature Communications]

Neuromodulatory subcortical nucleus integrity is associated with white matter microstructure, tauopathy and *APOE* statusREVIEWER COMMENTS

Reviewer #1 (Remarks to the Author):

Wearn et al use multiparametric quantitative magnetic resonance imaging to determine integrity measures of the isodendritic core, and how these relate to whole brain integrity. The cohort is 133 people with a first degree relative with Alzheimer's disease. Results identified two covariance patterns between isodendritic core neurons and the rest of the brain, with the second pattern associated with CSF-tau levels; and with both patterns more prominent in APOE4 carriers – together, leading the authors to conclude that the isodendritic microstructure relationships they identified have relevance for Alzheimer's pathology. This is a well conducted study, producing some novel results that would be of interest to an ageing/neurodegeneration readership.

- Atlases are used to define 3 out of the 4 isodendritic nuclei; however the dorsal raphe ROI is created using placement of an anatomically informed sphere. To keep consistency with the other nuclei I wondered why an available atlas wasn't used here, for example <https://www.nmr.mgh.harvard.edu/resources/aan-atlas>

- Why was the dopaminergic ventral tegmental area selected? Instead of substantia nigra which has been included in previous Alzheimer's work on the isodendritic core (e.g., <https://pubmed.ncbi.nlm.nih.gov/25720408/>). Based on morphology I suspect both would be considered part of the isodendritic core. Given the authors selected one over the other, it would be helpful to have justification for that.

- In a few places this point is highlighted: "We also highlight a particular vulnerability of dopaminergic and cholinergic nuclei in APOE4 carriers." And furthermore: "Apolipoprotein-E4 carriers expressed both patterns more strongly, and exclusively displayed variation in cholinergic and dopaminergic IdC nuclei." It wasn't clear to me where this result/interpretation comes from. Is the suggestion that the dopaminergic and cholinergic are the main drivers of variation in the APOE4-associated pattern? Which doesn't seem to equate with a selective vulnerability per se, and from Figure 6 those nuclei don't appear to vary selectively with the overall pattern.

Reviewer #2 (Remarks to the Author):

This study aimed to further our understanding of early neuropathological changes in Alzheimer's disease (AD) by investigating the relationship between i. the integrity of subcortical nuclei in the isodendritic core (IdC) (locus coeruleus, dorsal raphe, ventral tegmental area, and nucleus basalis of Meynert), measured by average values of proton density (PD), magnetization transfer saturation (MTsat), R1 and R2* from these ROIs and ii. whole white matter integrity measured with NODDI indices of neurite density, orientation dispersion, and free water in a sample of 133 older adults with a family history of AD. Using partial least square analysis, two covariance patterns were identified: the first relating IdC integrity to neurite density in whole-brain white matter and the second to orientation dispersion in the corpus callosum and the cingulum bundle, with latter being associated with CSF concentrations of pTau. Further the study found these patterns to be stronger expressed in APOE4+ carriers versus non-carriers. The authors conclude that variations in IdC integrity are reflected in white matter microstructure, notably in tracts susceptible to AD.

This is a complex study that addresses several important questions with regards to the role of IdC nuclei and white matter degeneration early in the pathological AD process as well as modulated effect of APOE4 genotype, the largest genetic risk factor for AD and other neurodegenerative diseases. The strength of the study lies in the combination of these variables. I have the following queries/comments:

- Many complex correlational analyses were conducted to reveal association pattern between 4 multi-parametric indices from 4 IdC ROIs and 3 NODDI indices from whole brain white matter voxels, between these patterns and CSF tau and amyloid measures, and separately for APOE4+ an

APOE4- groups. Significance of individual IdC measurements was assessed with 95% bootstrap confidence intervals roughly equivalent to $p < 0.05$. From Figures 2-4 it can be seen that the correlations between IdC measurements and the first and second voxelwise NODDI patterns as well as CSF biomarkers were quite small for the whole sample (most fall below 0.3). I could not work out if any kind of multiple testing correction was applied within and/or across the various analyses. The concern is that such small effects would probably not survive.

- Some of the findings are surprising and require further discussion and explanation. For instance, the first covariance pattern found associations with whole brain white matter except for prominent white matter bundles of the corpus callosum and cingulum. If this pattern reflects a general relationship between IdC and whole brain white matter integrity why is the relationship not significant for voxels along these bundles?
- The second pattern showed correlations with cingulum and CC labelled as limbic tracts susceptible to AD but not with the fornix tract. This is surprising given that the fornix is the main hippocampal white matter pathway that together with the parahippocampal cingulum is known to be disproportionately affected by aging and AD degeneration?
- The second pattern between IdC and white matter measures was mainly driven by R1, MTsat, and R2. This pattern was found to be related to CSF tau measures. However, Figure 2 shows that only R1 in the DR correlated with tau while R1 in LC correlated with amyloid. This perhaps suggests that the association between pattern 2 and CSF tau might be driven by the white matter NODDI indices rather than IdC differences? Was this checked?
- The authors acknowledge a number of limitations of the study including CSF partial volume artefacts in the IdC multi-parametric measurements that were acquired at 1mm³. Given that these measurements were not investigated across the whole brain it is unclear why the acquisition resolution was not optimized for brain stem/ midbrain regions.
- R1, MTSat and R2* were all interpreted as reflecting myelination in IdC regions (although this remains speculative). Given this interpretation it is unclear why NODDI metrics were investigated in whole brain white matter rather than the multi-parametric measurements.
- APOE4+ effects were interpreted in terms of its role in lipid metabolism but APOE4 has many functions and has also been associated with increased neuroinflammation and vascular impairment. It would be helpful to include information about the distribution of APOE alleles in the sample. 50 APOE4+ carriers seems quite a high proportion. Do the authors have any explanation for this?
- To conclude, although the study has used sophisticated analyses to reveal interesting associations between variation IdC integrity and brain white matter as well as tau CSF biomarkers and APOE genotype, the causality and the underpinning mechanisms of these correlations remain unclear.

Response to Referees

We would like to thank both reviewers for their thoughtful and constructive comments on our manuscript. Both reviewers raised important questions about the paper to which we are delighted to have the opportunity to respond in order to clarify our methods and findings.

Reviewer #1 (Remarks to the Author):

Reviewer 1, Summary and Assessment: *Wearn et al use multiparametric quantitative magnetic resonance imaging to determine integrity measures of the isodendritic core, and how these relate to whole brain integrity. The cohort is 133 people with a first degree relative with Alzheimer's disease. Results identified two covariance patterns between isodendritic core neurons and the rest of the brain, with the second pattern associated with CSF-tau levels; and with both patterns more prominent in APOE4 carriers – together, leading the authors to conclude that the isodendritic microstructure relationships they identified have relevance for Alzheimer's pathology. This is a well conducted study, producing some novel results that would be of interest to an ageing/neurodegeneration readership.*

Reviewer 1, Comment 1: *Atlases are used to define 3 out of the 4 isodendritic nuclei; however the dorsal raphe ROI is created using placement of an anatomically informed sphere. To keep consistency with the other nuclei I wondered why an available atlas wasn't used here, for example <https://www.nmr.mgh.harvard.edu/resources/aan-atlas>*

Response:

The decision to make our own ROI for dorsal raphe was not one made lightly, and we thank Reviewer 1 for highlighting the opportunity to justify and clarify this decision. We have added the following text to the 'Regions of Interest' Methods subsection:

The DR mask was manually defined. Although publicly available masks of DR do exist^{91–93}, we observed large discrepancies between their spatial profiles, with little overlap between them in some cases. For example, the mask of Bianciardi et al.⁹² (thresholded at 50%) extends considerably more rostrally, up to the level of the cerebral aqueduct and superior colliculus, than those of Levinson et al.⁹¹ (thresholded at 50%) and Edlow et al.⁹³ (Supplementary Fig. 1). These discrepancies may be due to biases toward different DR subnuclei, resulting from methodological differences in how the region is localized (explained further in supplemental information). Given these discrepancies, we decided to leverage the BigBrain histological atlas⁹⁴. This atlas allows for visualization of the serotonergic cell bodies

of the DR, and to directly translate their location to MNI coordinates thanks to a previously published precise registration pipeline⁹⁵. Combining this information with histological descriptions of DR^{96,97} and the Allen Human Brain Atlas⁹⁸, a spherical ROI (3mm radius) was drawn around MNI coordinates x: 0, y: -30, z: -13. Any edge voxels close to the cerebral aqueduct of periaqueductal grey were trimmed leaving a roughly spherical ROI of volume $\sim 32\text{mm}^3$. This position was chosen to align with the supratrochlear subnucleus of DR where neurofibrillary tangles have been identified in asymptomatic stages of Alzheimer's disease.¹⁴ Our mask is available online (see ref⁹⁹).

We also added further justification and an additional figure to supplemental information.

“Discrepancies in Dorsal Raphe localization

While validated atlases were used to define locus coeruleus, ventral tegmental area and nucleus basalis of Meynert, we elected in this study to use a manually defined mask of the dorsal raphe. Although publicly available masks of dorsal raphe do exist⁶⁻⁸, we observed large discrepancies between their spatial profiles, with little overlap between them in some cases. For example, the mask of Bianciardi et al.⁷ (thresholded at 50%) extends considerably more rostrally, up to the level of the cerebral aqueduct and superior colliculus, than those of Levinson et al.⁶ (thresholded at 50%) and Edlow et al.⁸ (Supplementary Fig. 1). Other studies examining the dorsal raphe define a $\sim 150\text{mm}^3$ region centered on MNI coordinates x: 0, y: -27 / -31, z: -9^{3,9,10}. This ROI, defined in relation to histological sources, is then refined in relation to serotonin transporter (5-TT) PET images, which highlight serotonergic neurons in the dorsal and median raphe nuclei. This places the location of dorsal raphe ROI fairly rostral, in line with the cerebral aqueduct. This is similar to the mask of Bianciardi et al.⁷, but does not overlap with the masks of Levinson et al.⁶ and Edlow et al.⁸.

These discrepancies may arise as a result of methodology. Studies which place DR more rostrally tend to rely on direct histological reference and/or co-localization with 5-TT PET and may therefore be more localized to the serotonergic neurons in the rostral supratrochlear nucleus of the dorsal raphe, assuming caution is taken to avoid accidentally including the serotonin-rich median raphe nucleus¹¹). This supratrochlear nucleus, critically, is where neurofibrillary tangles have been identified¹², and is the particular subregion of interest for the present study. Studies which localize dorsal raphe more caudally^{6,8} tend to rely on diffusion imaging and may be more sensitive to identifying cells with more densely packed afferent and efferent fibres, regardless of the subnucleus. Diffusion-based methods may be more sensitive to localizing more caudal sections of DR, at the expense of the more rostral subnuclei.”

Supplementary Fig. 1 | Location of dorsal raphe ROI in MNI space. A: Coronal slice of BigBrain ($y=-8.25\text{mm}$). The 'wings' of the supratrochlear nucleus of dorsal raphe are clearly visible, highlights my red arrow for right hemisphere. B: Coronal slice of MNI-space BigBrain (MNI-space $y=-30.5$). The view is analogous to that in panel A, with the same supratrochlear dorsal raphe 'wing' highlighted. C: Sagittal view of MNI-space BigBrain ($x=1$), with arrow highlighting the right-hemisphere dorsal raphe 'wing', visible as a dark spot in this view, in line with the inferior colliculus. D: The same view as in C, but with a red circle highlighting our manually-defined dorsal raphe ROI. E: The same view as in D, but with the extra overlays of Bianciardi 'Brainstem Navigator' dorsal raphe ROI (blue, thresholded at 50%) and the Levinson et al. dorsal raphe ROI (yellow, thresholded at 50%). F: The same sagittal view as in E, but the image shows T1-weighted MNI-template instead of BigBrain.

Reviewer 1, Comment 2: Why was the dopaminergic ventral tegmental area selected? Instead of substantia nigra which has been included in previous

Alzheimer's work on the isodendritic core

(e.g., <https://pubmed.ncbi.nlm.nih.gov/25720408/>). Based on morphology I suspect both would be considered part of the isodendritic core. Given the authors selected one over the other, it would be helpful to have justification for that.

Response:

Thank you for highlighting this point, we agree that this needs to be justified as both nuclei are often discussed in the context of Alzheimer's disease. We have added the following discussion to the 'Regions of Interest' section in Methods:

“Substantia nigra rarely exhibits significant atrophy compared to the rest of the isodendritic core and there is less evidence for early tauopathy and degeneration in substantia nigra than VTA (see ¹ for a review). In contrast, dopaminergic neuronal loss has been localised specifically to VTA (while absent in substantia nigra) in a mouse model of Alzheimer's disease ². Furthermore, in vivo evidence in humans suggests that above and beyond any association with substantia nigra, VTA exhibits lower T1-weighted signal, indicating microstructural changes, in individuals who experience more cognitive decline compared to cognitively stable individuals ³. Finally, in the earliest stages of Alzheimer's disease, there is evidence to suggest that substantia nigra is more susceptible to alpha-synuclein deposition than neurofibrillary tangles ⁴, and therefore may represent different disease mechanisms or co-pathology. In support of this, substantia nigral pathology is more closely related to the presence of extrapyramidal motor symptoms such as dyskinesia and Parkinsonism, which tend to become more prevalent in later stages of disease ⁴. In contrast, VTA atrophy is more closely associated with cognition deficits ⁵. For these reasons we focused on VTA over substantia nigra, however the individual roles of both nuclei in early and late stages of Alzheimer's disease, and their relation to comorbidities requires considerably more work to untangle.”

Reviewer 1, Comment 3: *In a few places this point is highlighted: “We also highlight a particular vulnerability of dopaminergic and cholinergic nuclei in APOE4 carriers.” And furthermore: “Apolipoprotein-E4 carriers expressed both patterns more strongly, and exclusively displayed variation in cholinergic and dopaminergic IdC nuclei.” It wasn't clear to me where this result/interpretation comes from. Is the suggestion that the dopaminergic and cholinergic are the main drivers of variation in the APOE4-associated pattern? Which doesn't seem to equate with a selective vulnerability per se, and from Figure 6 those nuclei don't appear to vary selectively with the overall pattern.*

Response

Thank you for catching this – the way this conclusion is currently written is indeed confusing and failed to capture the core results of the manuscript. We apologize for the confusion and have now revised the main text.

The message we intended to convey was that the microstructure of all four nuclei appear to be related to white matter structure (of AD-associated regions) in APOE4 carriers (pattern 2), whereas in APOE4 non-carriers the relationship between isodendritic core microstructure and white matter is largely restricted to LC and DR (pattern 3). In other words, the dopaminergic (VTA) and cholinergic (NbM) nuclei are related to white matter specifically in APOE4 carriers, and not so much in APOE4 non-carriers.

We have modified the sentences highlighted and added some clarifications to the discussion to better reflect this conclusion:

Abstract: “Apolipoprotein-E4 carriers expressed both patterns more strongly than non-carriers”

Discussion:” This APOE4- pattern also highlighted microstructural covariance in the IdC restricted to LC and DR. In contrast, the APOE4+ pattern (LV2) involved all four IdC nuclei, suggesting APOE4- individuals may show relatively restricted microstructural variation within IdC compared to APOE4+ carriers. LC and DR also interact with the APOE4 genotype (LV4): integrity in LC and DR is associated with a distributed pattern of voxels in limbic tracts and peripheral white matter in opposite directions for each genotype. IdC nuclei may be differentially vulnerable in Alzheimer’s pathology depending on APOE4 genotype, with VTA and NbM expressing greater APOE4-related risk than LC and DR. Further study of these nuclei and neuromodulatory systems, particularly dopaminergic and cholinergic systems, may be critical to understanding the mechanisms behind APOE4-related Alzheimer’s risk.”

Conclusion: “We also highlight that dopaminergic and cholinergic nuclei are specifically vulnerable in APOE4 carriers compared to non-carriers”

Reviewer #2 (Remarks to the Author):

Reviewer 2, Summary and Assessment: *This study aimed to further our understanding of early neuropathological changes in Alzheimer's disease (AD) by investigating the relationship between i. the integrity of subcortical nuclei in the isodendritic core (IdC) (locus coeruleus, dorsal raphe, ventral tegmental area, and nucleus basalis of Meynert), measured by average values of proton density (PD), magnetization transfer saturation (MTsat), R1 and R2* from these ROIs and ii. whole white matter integrity measured with NODDI indices of neurite density, orientation dispersion, and free water in a sample of 133 older adults with a family history of AD. Using partial least square analysis, two covariance patterns were identified: the first relating IdC integrity to neurite density in whole-brain white matter and the second to orientation dispersion in the corpus callosum and the cingulum bundle, with latter being associated with CSF concentrations of pTau. Further the study found these patterns to be stronger expressed in APOE4+ carriers versus non-carriers. The authors conclude that variations in IdC integrity are reflected in white matter microstructure, notably in tracts susceptible to AD.*

This is a complex study that addresses several important questions with regards to the role of IdC nuclei and white matter degeneration early in the pathological AD process as well as modulated effect of APOE4 genotype, the largest genetic risk factor for AD and other neurodegenerative diseases. The strength of the study lies in the combination of these variables. I have the following queries/comments:

Response:

We thank the reviewer for their constructive assessment and fulsome comprehension of the core aims of our work.

Reviewer 2, Comment 1: *Many complex correlational analyses were conducted to reveal association pattern between 4 multi-parametric indices from 4 IdC ROIs and 3 NODDI indices from whole brain white matter voxels, between these patterns and CSF tau and amyloid measures, and separately for APOE4+ an APOE4- groups. Significance of individual IdC measurements was assessed with 95% bootstrap confidence intervals roughly equivalent to $p < 0.05$. From Figures 2-4 it can be seen that the correlations between IdC measurements and the first and second voxelwise NODDI patterns as well as CSF biomarkers were quite small for the whole sample (most fall below 0.3). I could not work out if any kind of multiple testing correction was applied within and/or across the various analyses. The concern is that such small effects would probably not survive.*

Response:

Thank you for highlighting this, we agree that it is extremely important to ensure multiple statistical tests are appropriately thresholded to reduce the rate of false-positive effects. In this study, to avoid having many independent univariate tests over

which we would be required to perform a correction for multiple tests, we chose to leverage a small number of multivariate analyses. A key strength of partial least squares (PLS) as a multivariate analysis technique is that multiple testing correction is not required because the analysis is performed in a single analytic step across all observations. While many correlation values are shown for each latent variable, they are not independent tests, as they are derived from a single model. Therefore, correction for multiple comparisons across each of these variables is not warranted. We have added the following text to the methods section to highlight this:

“PLS facilitates the examination of covariance between numerous variables in a single model and a single analytic step, without necessitating the need for multiple-comparisons correction”

Multiple comparisons correction may be justified for the two linear regressions comparing PLS scores and CSF pTau (Fig 5), however our result of $p=0.004$, even with a conservative Bonferroni correction, would still be considered statistically significant ($p_{adj}=0.008$) at our critical alpha level (now also stated in methods). This has now been changed in the results section, and described in the methods section:

Results: **“No association was observed with NDI in LV1 ($b = 0.146$, $t = 1.45$, $p_{adj} = .300$, **Error! Reference source not found.A**). However, we observed a significant association with ODI in LV2 ($b = 0.295$, $t = 2.93$, $p_{adj} = .008$, **Error! Reference source not found.B**)”**

Methods: **“P-values from these two tests were Bonferroni corrected for multiple comparisons (shown as p_{adj}). All statistical tests were two-tailed ($\alpha = 0.05$).”**

The tests summarised in Figure 2 are provided as a general description of the dataset, and not direct hypothesis testing (as stated in the legend), so we do not feel they need to be corrected for multiple comparisons.

Reviewer 2, Comment 2: *Some of the findings are surprising and require further discussion and explanation. For instance, the first covariance pattern found associations with whole brain white matter except for prominent white matter bundles of the corpus callosum and cingulum. If this pattern reflects a general relationship between IdC and whole brain white matter integrity why is the relationship not significant for voxels along these bundles?*

Response:

It is indeed interesting that midline white matter tracts (including corpus callosum) do not follow the same relationship with IdC as the rest of the more distal cerebral white matter that is described by LV1 of our PLS analysis. We can speculate that the tightly bundled, highly directional microstructure of corpus callosum allows for less variation in neurite density (described by LV1) while allowing for variation in orientation dispersion (described primarily by LV2), particularly as a function of IdC integrity. The exact reasons are not currently known given the existing methods. We believe this is

a worthy topic for future study. We are grateful to the reviewer for highlighting this. We have added the following text to the discussion:

“A notable exception from this global pattern are voxels within midline regions such as corpus callosum, indicating a different relationship with isodendritic core than the rest of cerebral white matter. The tightly bundled, highly directional microstructure of corpus callosum may allow for less variation in neurite density (and therefore is not described by pattern 1) while allowing for variation in orientation dispersion (and is therefore described by pattern 2). Further exploration and validation of this relationship would be an interesting topic for future study.”

Reviewer 2, Comment 3: *The second pattern showed correlations with cingulum and CC labelled as limbic tracts susceptible to AD but not with the fornix tract. This is surprising given that the fornix is the main hippocampal white matter pathway that together with the parahippocampal cingulum is known to be disproportionately affected by aging and AD degeneration?*

Response:

We appreciate the reviewer's observations regarding the fornix. Indeed, given its sensitivity to aging and Alzheimer's disease, its exclusion warrants clarification. As you will no doubt be aware, the fornix is a relatively thin structure that is surrounded by non-white matter tissue types, including CSF. As ventricle size was highly variable within our older population, we wanted to minimize risk of partial voluming from ventricles in our assessment of white matter microstructure, and therefore employed a strict threshold (95%) on our probabilistic white matter mask. This threshold excluded fornix given its adjacency to the CSF. Our 2mm isotropic diffusion imaging resolution required conservative voxel selection to ensure reliable data, precluding a focused analysis on the fornix without robust *a priori* hypotheses.

We have now discussed this as a limitation of our study in the following text:

“To ensure we were only exploring patterns within white matter, and not neighbouring CSF or grey matter regions, we strictly thresholded our white matter mask to include only voxels identified as white matter for at least 95% of subjects. This criterion prioritized voxel specificity at the cost of sensitivity, and therefore excluded very peripheral white matter areas as well as thinner isolated structures. Unfortunately, this meant excluding certain regions of potential interest such as the fornix, which has been previously implicated in aging and Alzheimer's disease. Future studies specifically examining fornix are of great interest and relevance to this field.”

Reviewer 2, Comment 4: *The second pattern between IdC and white matter measures was mainly driven by R1, MTsat, and R2. This pattern was found to be related to CSF tau measures. However, Figure 2 shows that only R1 in the DR correlated with tau while R1 in LC correlated with amyloid. This perhaps suggests*

that the association between pattern 2 and CSF tau might be driven by the white matter NODDI indices rather than IdC differences? Was this checked?

Response:

We are grateful to the reviewer for this insight and opportunity for clarification. The brainscores correlated with CSF pTau in Figure 5 are the result of the dot product between the group-level salience image and the individual subject NODDI images. These scores provide a participant level score for the strength of the association between the voxelwise NODDI metrics and IdC microstructure. Our observation that the LV2 brainscores (linking IdC microstructure with white matter microstructure) relate to pTau can be interpreted as the identified white matter microstructure being related to CSF pTau. Given this conclusion, we would indeed expect a direct relationship between pTau and NODDI measures across a pattern of voxels with a distribution that at least partially overlaps with the pattern displayed by LV2.

Supporting this, our additional PLS analysis reveals a significant correlation between log-pTau and age-adjusted NODDI indices, revealing a pattern similar to LV2. The pattern includes regions of corona radiata where pTau is positively related to ODI and negatively related to NDI and FW (shown in yellow). The pattern also includes regions of cingulum bundle where pTau is negatively related to ODI and positively related to NDI and FW (shown in blue). We have added the results of this analysis to the Supplementary Materials and signposted it appropriately in the methods section of the manuscript.

Methods: “We also ran an additional PLS correlation analysis between CSF pTau and NODDI maps to explore direct relationships between white matter microstructure and Alzheimer’s disease pathology (see Supplementary Materials).”

Supplemental: “**Testing voxelwise relationship between CSF pTau and NODDI measures**

We found that participant brainscores for LV2 of the primary analysis (Fig. 3), which quantitatively express the relationship between IdC and white matter microstructure, was related to CSF pTau. This was further validated in a supplemental analysis where we directly assessed the association between white matter microstructure and CSF pTau.

PLS analysis revealed a significant correlation between log-pTau and age-adjusted NODDI indices, revealing a single significant LV: 43% crossblock covariance explained, $p=0.031$ (Supplemental Fig. 2). The pattern included regions of corona radiata where pTau is positively related to ODI and negatively related to NDI and FW (shown in yellow). The pattern also includes regions of cingulum bundle and longitudinal fasciculus as well as temporal lobe white matter where pTau is negatively related to ODI and positively related to NDI and FW (shown in blue).

These supplementary results reveal that the spatial topography relating IdC microstructure with white matter microstructure (main analysis LV2), corresponds with the spatial topography relating CSF pTau to white matter microstructure (Supplemental Fig. 2C). This correspondence supports the interpretation that LV2 is a pathological pattern related to AD.

Supplemental Fig. 2 | PLS correlation results between log-pTau CSF concentration and NODDI measures in white matter. A) The strength and direction of the relationship between pTau and each respective NODDI map. Error bars show 95% confidence intervals. B) The yellow voxels indicate a positive relationship with the pattern shown in (B). The blue voxels indicate a negative relationship. Only voxels with a bootstrap ratio of $>|2|$ are colored. C) The spatial overlap of patterns shown in panel B and LV2 of the primary analysis (figure 4 of main manuscript). NDI = Neurite Density Index, ODI = Orientation Dispersion Index, FW = Free water fraction

Reviewer 2, Comment 5: The authors acknowledge a number of limitations of the study including CSF partial volume artefacts in the IdC multi-parametric measurements that were acquired at 1mm3. Given that these measurements were not investigated across the whole brain it is unclear why the acquisition resolution was not optimized for brain stem/ midbrain regions.

Response:

We thank the reviewer for raising this point. We considered many options for optimising the multiparametric mapping sequence to brainstem/midbrain imaging, including limiting the field-of-view (FOV) to a localised slab instead of collecting whole brain images. A smaller FOV (e.g. restricted to a brainstem/midbrain slab) would actually lead to a reduction in signal-to-noise ratio (SNR) compared to a whole-brain

field of view, albeit at a faster scan time, as SNR is proportional to the square root of the number of data points acquired. Utilising this extra time to acquire more images over which to average would provide no extra SNR compared to acquiring a larger FOV. In real terms, doubling the FOV would increase SNR by a factor of four, whereas doubling the number of averages would increase SNR by a factor of only two. Further increasing the scan time in order to provide better SNR through additional averages was also not an appropriate option due to the increased risk of motion in our older adult cohort. Acquiring whole-brain images had the added benefit of enabling these scans to be used more flexibly by the wider community upon release as an open-access dataset. We acknowledge the benefits of higher-resolution imaging for the nuclei of the isodendritic core and consider it a valuable direction for future investigations to better understand these regions and their relationship to aging and neurodegenerative disease. We have added the following text to the limitations paragraph on the topic of scan resolution:

“Limiting the MPM sequence field-of-view from whole brain to a brainstem/midbrain slab would significantly decrease signal-to-noise, albeit at a faster scan time. Utilizing this extra time to average additional image acquisitions would not provide any appreciable benefit to signal-to-noise compared to fewer averages using a whole brain acquisition. Further increasing the scan time in order to provide better signal to noise was also not an appropriate option due to the increased risk of motion in our older adult cohort. Acquiring whole-brain images had the added benefit of enabling these scans to be used more flexibly by the wider community upon release as an open-access dataset. We acknowledge the benefits of higher-resolution imaging for the nuclei of the isodendritic core and consider it a valuable direction for future investigations to better understand these regions and their relationship to aging and neurodegenerative disease.”

Reviewer 2, Comment 6: *R1, MTSat and R2* were all interpreted as reflecting myelination in IdC regions (although this remains speculative). Given this interpretation it is unclear why NODDI metrics were investigated in whole brain white matter rather than the multi-parametric measurements.*

Response: We wholeheartedly agree that MPM measures of the whole brain could provide valuable insights into brain health. Indeed, we did consider additionally including the MPM measures as measures of white matter microstructure as well as NODDI parameters, however further integrating MPM measures into the white matter matrices significantly complicated interpretation of the already complex findings. The NODDI model offers a robust framework for characterizing white matter microstructure with specific regard to fibre orientation and complexity, which was pivotal in elucidating the results of our study, particularly those related to pattern 2. Therefore, in order to balance model completeness with model interpretability, we

focused on the NODDI model to inform our conclusions. However, we have ongoing projects in our group using the MPM parameters to assess microstructure broadly across the whole brain.

We have modified the manuscript to highlight this point:

“We did not examine MPM-derived measures across the whole-brain white matter in order to focus our findings within the context of the NODDI model. This approach enabled us to more specifically examine relationships between the IdC and white matter microstructure.”

Reviewer 2, Comment 7: *APOE4+ effects were interpreted in terms of its role in lipid metabolism but APOE4 has many functions and has also been associated with increased neuroinflammation and vascular impairment. It would be helpful to include information about the distribution of APOE alleles in the sample. 50 APOE4+ carriers seems quite a high proportion. Do the authors have any explanation for this?*

Response: Thank you for these points, we agree that other potential mechanisms for APOE4 vulnerability on IdC should be mentioned. We have added a discussion of the importance of inflammation and vascular dysfunction in the context of APOE genotype, and how it relates to IdC vulnerability to the discussion:

“APOE4 may also mediate disease through altered inflammatory states and vascular dysfunction⁷². Damaged cerebral vasculature or increased presence of inflammatory mediators may disproportionately affect the isodendritic core given the greater metabolic demands of the long and highly arborized neurons⁵⁻⁷. Future studies should address the APOE4 genotype’s impact on myelination, inflammation, and vascular dysfunction in relation to vulnerability of IdC projections in humans.”

Regarding the high proportion of APOE4 carriers, our data came from the PREVENT-AD study, an ongoing study of cognitively healthy individuals with a first-degree family history of Alzheimer’s disease. Individuals for this study have been specifically recruited for familial risk, which explains the higher proportion of APOE4 carriers compared to a general population sample. A short explanation has been added to the manuscript:

“This is a greater proportion of carriers than would be found in the general population, explained by the fact that these individuals have been specifically recruited to have familial risk (first-degree family history) for Alzheimer’s disease.”

Reviewer 2, Comment 8: *To conclude, although the study has used sophisticated analyses to reveal interesting associations between variation IdC integrity and brain white matter as well as tau CSF biomarkers and APOE genotype, the causality and the underpinning mechanisms of these correlations remain unclear.*

Response:

This is absolutely correct and is a limitation of this study as with any correlational study. We have been careful to avoid causative language throughout the study but have now also added an additional note to highlight this:

“Finally, as with any correlational study, presence or direction of causality cannot be determined from these findings. Longitudinal and intervention studies should explore these regions and their relationship with aging and neurodegenerative disease with the specific aim to determine causative mechanisms.”

Other changes:

In response to a comment from a co-author, we have modified slightly the language on inclusion and ethics and added full cohort-specific acknowledgments and detailed recruitment strategy information to the Supplemental Materials.

Inclusion & Ethics statement:

Research ethics for the study was obtained from the Research Ethics Board of the Faculty of Medicine and Health Science at McGill University **and/or the Comité d'éthique de la recherche du CIUSSS de l'ouest de l'île de Montréal**. Participants provided written informed consent. All participants included in this study were recruited as part of PREVENT-AD recruitment protocols in **2011-2017**.

Acknowledgements:

A full list of PREVENT-AD authors and collaborators can be found here and further information about the program can be found here. Full acknowledgments can be found in Supplemental Materials.

Supplemental Materials Acknowledgements:

About PREVENT-AD

Full Acknowledgments

PREVENT-AD was launched in 2011 as a \$13.5 million 7-year public-private partnership using funds provided by McGill University, the Fonds de Recherche du Québec-Santé (FRQS), an unrestricted research grant from Pfizer Canada, the Levesque Foundation, the Douglas Hospital Research Centre and Foundation, the Government of Canada, the Canada Fund for Innovation, the Canadian Institutes for Health Research, the Alzheimer Society of Canada and the Alzheimer Association. Private sector contributions are facilitated by the Development Office of the McGill University Faculty of Medicine and by the Douglas Hospital Research Centre Foundation (www.douglas.qc.ca).

The primary goal of PREVENT-AD is to test whether serial determination of multi-modal biomarkers of Alzheimer's disease may be measured and used in pre-symptomatic persons at high risk of subsequent AD dementia to trace the progression of the disease process and to measure effects of any potentially preventive treatment interventions. This work is intended to provide preliminary data regarding the probably efficacy and safety of potential new treatments for prevent of AD dementia.

The Founders of the program were John C. S. Breitner, MD, MPH, Judes Poirier, PhD, and Pierre Etienne, MD, Douglas Hospital Research Centre and Faculty of Medicine, McGill University, Montreal, QC, Canada. Program Current Director is Sylvia Villeneuve, PhD, the Co-Director is Judes Poirier, PhD, and the Study Coordinator is Jennifer Tremblay-Mercier, MSc. PREVENT-AD is the result of efforts of many other co-investigators from a range of academic institutions and private corporations, as well as an extraordinarily dedicated and talented clinical and technical assistant staff, students, and postdoctoral fellows. Subjects are recruited

from the greater Montreal area and more distant locations in Quebec. For up-to-date information see https://prevent-alzheimer.net/?page_id=42&lang=en

Recruitment Strategy

To bring this population into our program, the main strategy was to send informative flyers about the PREVENT-AD program, in a bag called “publisac” containing different kind of publicity, weekly delivered to the Quebec population. People were invited to contact us by telephone or visit our web site (www.prevent-alzheimer.ca) if they were 55 or older and if they had a parent, brother or sister who has/had Alzheimer’s disease. To date, 250,000 flyers were sent in 65 different areas Montreal, QC, CAN. The areas were chosen based on demographic statistics i.e. in areas with higher proportion of 55 years old or older, compared to the provincial average. People were also reached by various media coverage done on the Stop-AD Center and on principal investigators of the Center (television, radio, newspapers). At a smaller level, the program was presented in different academic institutions and nursing homes and flyers were distributed to medical professionals and general population in various occasions.

REVIEWERS' COMMENTS

Reviewer #1 (Remarks to the Author):

The authors thoroughly addressed my comments. (adding the rationale for the raphe mask selection and the current state of available masks was a particularly useful addition -- thank you).

Reviewer #2 (Remarks to the Author):

The authors have addressed the points I raised in response to the original manuscript and I have no further comments.